# OptimSyn: Influence-Guided Rubrics Optimization for Synthetic Data Generation

**Zhiting Fan**[1,2], **Ruizhe Chen**[1], **Tianxiang Hu**[1], **Ru Peng**[1], **Zenan Huang**[1,2], **Haokai Xu**[1]
**Yixin Chen**[1], **Jian Wu**[1], **Junbo Zhao**[1,2], **Zuozhu Liu**[1,3] [*]
[1]Zhejiang University, [2]Inclusion AI, Ant Group,
[3]Zhejiang Key Laboratory of Medical Imaging Artificial Intelligence
{zhiting.23, zuozhuliu}@intl.zju.edu.cn

## Abstract

Large language models (LLMs) achieve strong downstream performance largely due to abundant supervised fine-tuning (SFT) data that imparts problem-solving capabilities. However, as applications expand, high-quality SFT data in knowledge-intensive domains (e.g., humanities and social sciences, medicine, law, finance) is exceedingly scarce: expert curation is costly, privacy constraints are strict, and label consistency is hard to guarantee. Recent work turns to synthetic data, typically prompting a generator over domain documents and filtering with handcrafted rubrics. Yet, rubric design is expert-dependent and rarely transfers across domains; moreover, prevalent heuristic optimization follows a brittle loop (*write rubric → synthesize → train → inspect → guess tweaks*) that lacks reliable, quantitative feedback about a rubric's true contribution to downstream performance. We argue for assessing synthetic data quality through its *training utility* on the target model, using this feedback to guide data generation. Inspired by classic influence estimations, we repurpose an optimizer-aware estimator that uses gradient information to quantify each synthetic sample's contribution to the objective of a given target model on specific tasks. Our analysis reveals a gap: although synthetic and real samples may be close in embedding space, their *influence* on learning can differ substantially. Building on this insight, we propose an *optimization-based synthetic data framework* that adapts rubrics with *target-model feedback*. Instead of manually engineering domain rubrics, we supply lightweight guiding text and delegate rubric generation to a rubric-specialized model conditioned on the task; crucially, we employ influence score as reward and optimize the rubric generator with reinforcement learning. Empirically, the framework yields consistent gains across domains (HSS and health), target models (e.g., Qwen and Llama families), and data generators, demonstrating broad generalization and engineering portability without task-specific tuning.

## 1 Introduction

Large language models (LLMs) now excel across a wide range of downstream tasks (Achiam et al., 2023; Comanici et al., 2025; Guo et al., 2025), due in no small part to training on vast and diverse corpora. In particular, supervised fine-tuning (SFT) has endowed LLMs with strong instruction-following and problem-solving capabilities (Peng et al., 2023; Maeng et al., 2017). However, as LLMs are deployed in increasingly specialized scenarios, *high-quality, real* SFT data has become acutely scarce—especially in knowledge-intensive verticals such as the humanities and social sciences, medicine, law, and finance. The barriers are both practical and structural: domain expertise is costly and limited, privacy constraints are stringent, and large-scale manual annotation is difficult to standardize, expensive to procure, and hard to keep consistent.

Synthetic data has recently emerged as a promising way to alleviate this bottleneck (Wang et al., 2022; Maeng et al., 2017). Most current pipelines bootstrap from raw materials (e.g., documents), query a teacher model to produce question–answer pairs, and then filter (or steer) generations with

---

[*]Corresponding Author

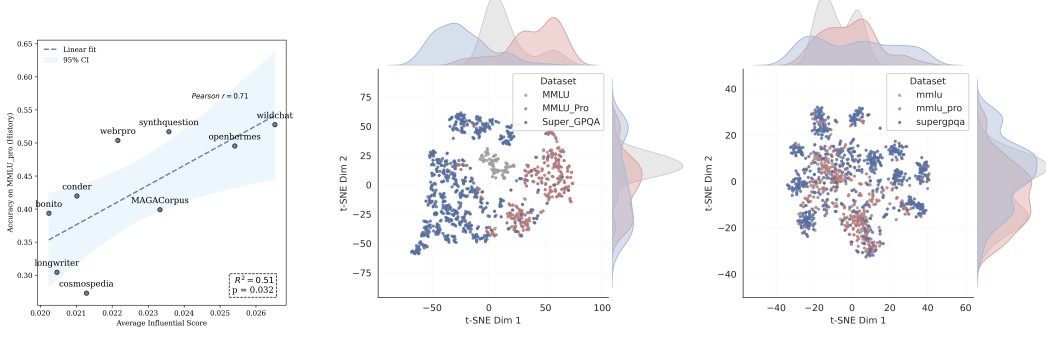

(a) Influences w.r.t. gains.    (b) Distributions in *embedding* space.    (c) Distributions in *gradient* space.

Figure 1: Visualization of influence score as an effective feature for assessing sample utility. (a) Relationship between validation influence scores and training performance. (b) Distribution of three validation sets in the embedding space of Qwen3-8B-Base. (c) Distribution of three validation sets in the loss–gradient embedding space, showing closer alignment and shared optimization directions.

pre-defined *rubrics* (rules or prompts) (Penedo et al., 2024; Jiang et al., 2025; Yue et al., 2024). Yet this paradigm faces two fundamental limitations. **(i) Limited transferability.** Rubric design is largely expert-driven and domain-specific; rubrics that work in one field often fail to generalize to another, undermining the universality of synthesis strategies. **(ii) Heuristic optimization.** The common loop—"handcraft rubrics → synthesize data → train model → inspect outcomes → propose revisions"—relies on tacit experience and provides little quantitative feedback. Human observers cannot reliably attribute downstream performance changes to specific rubric choices, making the process slow, brittle, and uncertain.

To address these limitations, we propose to directly quantify the *training utility* of each synthetic example for a given target model and task using a gradient-based influence estimator (Xia et al., 2024). Concretely, we approximate the contribution of a candidate QA pair to the validation objective under the same update rule used during fine-tuning (Adam), yielding an influence signal that is aligned with the actual optimization dynamics. Preliminary analyses (Figure 1) reveal a marked gap between representation proximity and training impact: synthetic and real samples can be close in embedding space yet diverge substantially in their estimated influence, which helps explain why seemingly on-distribution samples may still underperform during SFT. Moreover, dataset-level influence aggregates exhibit a strong positive correlation with held-out accuracy, validating influence as a reliable proxy for synthetic data quality. These observations motivate replacing heuristic rubric engineering with model-impact supervision grounded in gradients.

Building on this insight, we introduce a prompter (the policy model) optimization framework that treats rubric construction as a learnable component driven by target-model feedback. Starting from minimal guiding text, a rubric generator proposes seed document-specific rubrics; a generator model, conditioned on the seed document and rubric, synthesizes a QA pair; and the target model supplies a verifiable reward that combines lightweight validity checks with the optimizer-aware influence score. We then update the policy using GRPO algorithm (Guo et al., 2025), thereby closing the synthesis–training loop and explicitly maximizing expected downstream improvement. Empirically, this framework yields consistent gains across knowledge-intensive domains (the humanities and social sciences, and medicine and health), transfers across target model families and scales, and is robust to the choice of data generator, turning rubric design from brittle, expert-crafted heuristics into a portable, model-aligned optimization problem. The contributions of this paper are threefold:

1. **Optimizer-aware influence estimation.** We adapt classical influence-estimation ideas to modern synthetic-data pipelines and derive a practical, optimizer-aligned estimator of each synthetic sample's training utility.

2. **From heuristics to supervision.** We replace heuristic rubric engineering with a model-impact objective by training a *rubric generator* to maximize estimated downstream benefit (compatible with RL or gradient-based updates), yielding more effective synthetic data.

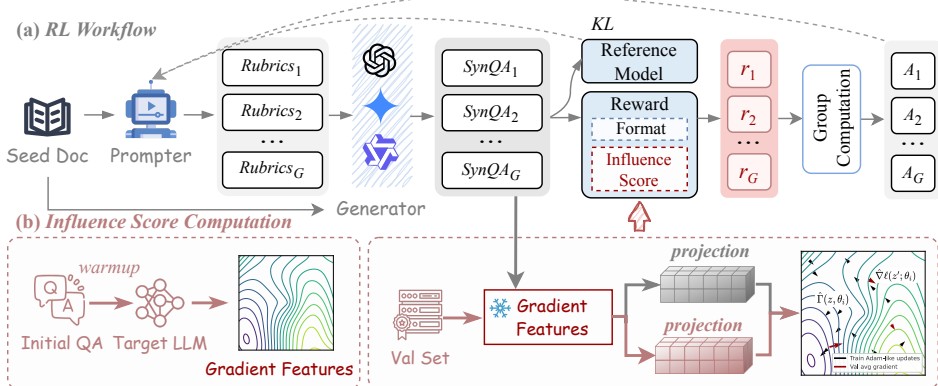

Figure 2: **The proposed RL framework for synthetic data generation with optimizer-aware rewards. (a) RL workflow.** Starting from a seed document, a *prompter* instantiates $G$ seed-specific rubrics, and a *generator* produces synthetic QA pair. **(b) Reward computation.** We employ the influence score for each synthetic QA pair as reward, followed by policy update.

3. **Transferable and efficient across domains.** Empirically, the learned rubrics are *model- and task-conditioned*, reducing reliance on domain expertise and enabling deployment in privacy-constrained, knowledge-intensive fields where manual curation is infeasible.

## 2 METHOD

### 2.1 OVERVIEW

Given a set of seed data $\mathcal{S} = \{S_i\}_{i=1}^N$, the goal of *data synthesis* is to construct a dataset containing synthetic question–answer pairs $\{(Q_i, A_i)\}_{i=1}^N$. Existing methods typically instruct a teacher model—often a strong external model (e.g., an API)—with carefully designed rubrics to generate such data. However, rubric-based synthesis is constrained by human intuition, highly domain-specific, and thus poorly transferable across tasks and models. To overcome these limitations, our framework employs a dedicated *rubric generator* that produces a targeted rubric $B_i$ for each seed datum $S_i$ and a specified target model. We then condition the teacher model on $(B_i, S_i)$ to generate a synthetic pair $(Q_i, A_i)$. In the following, we detail how we quantify the training utility of a synthetic pair $(Q_i, A_i)$ using optimizer-aware influence estimates and how this feedback drives the learning of the rubric generator.

### 2.2 ESTIMATING THE INFLUENCE OF SYNTHETIC DATA

While human-crafted rubrics appear reasonable, there is no guarantee that they actually lead to effective dataset generation. A promising yet underexplored alternative is to adopt a more principled generation process that does not rely solely on human intuition. Influence functions (Pruthi et al., 2020), which approximate training dynamics via first-order analysis, have been widely used to estimate the impact of individual training examples on held-out performance. Prior work (Xia et al., 2024) has shown their effectiveness in selecting data that benefits downstream tasks. This naturally motivates our use of influence functions as a principled guide for dataset generation.

**Preliminaries.** The classical influence function (Koh & Liang, 2017) quantifies how individual training points affect a model's parameters and, consequently, its predictions. TracIn (Pruthi et al., 2020) pursues the same goal with a scalable, first-order, trajectory-based estimator: it accumulates gradient inner products of training checkpoints to approximate influence. Because it requires only per-example gradients, learning rates, and saved checkpoints, TracIn is practical at LLM scale. Since contemporary LLMs are typically optimized with Adam (Adam et al., 2014), we adopt an Adam-compatible variant (Xia et al., 2024). Given a training sample $z$, its influence on an evaluation

---

**Algorithm 1** Influence-Guided Data Synthesis Framework

---

**Require:** Seed data set $\mathcal{S}$, Generator model $T$, Target model $M$, Policy model $\pi_\theta$, Validation set $\mathcal{D}_{\text{val}}$, Number of updates $K$, Rollout group size $G$, Reference model $\pi_{\text{ref}} \leftarrow \pi_\theta$;

1: **for** $k = 1$ to $K$ **do**
2:     Sample a mini-batch of seed data $\{S_i\}_{i=1}^B \sim \mathcal{X}$.
3:     **for** each seed data $S_i \in \mathcal{S}$ **do**
4:         Generate $G$ rubrics $B_{i,j} \sim \pi_\theta(\cdot | S_i)$ for $j = 1..G$
5:         **for** each rubric $B_{i,j}$ **do**
6:             Obtain $(Q_{i,j}, A_{i,j}) \leftarrow T(S_i, B_{i,j})$
7:             Estimate influence score $\text{IF}_{Adam}(Q, A)$ according to Eq. 1
8:             Compute trajectory reward $R(\tau) = \text{Valid}(Q, A) \cdot \text{IF}(Q, A) - \lambda(1 - \text{Valid}(Q, A))$
9:         **end for**
10:         Aggregate rewards and compute group-normalized advantages:
11:         $\hat{A}_{i,t} = \left(R(\tau_i) - \frac{1}{G}\sum_{j=1}^G R(\tau_j)\right) / \sqrt{\frac{1}{G}\sum_{j=1}^G \left(R(\tau_j) - \frac{1}{G}\sum_{k=1}^G R(\tau_k)\right)^2 + \delta}$
12:         Aggregate loss over tokens:
13:         $L(\theta) = -\frac{1}{G}\sum_{i=1}^G \frac{1}{|\tau_i|}\sum_{t=1}^{|\tau_i|} \min\left[r_{i,t}(\theta)\,\hat{A}_{i,t},\ \text{clip}(r_{i,t}(\theta))\,\hat{A}_{i,t}\right] - \beta\,\text{KL}(\pi_\theta \| \pi_{\text{ref}})$
14:         Update policy $\pi_\theta$
15:     **end for**
16: **end for**
17: **return** Optimized prompter $\pi_\theta^*$

---

sample $z'$ can be represented as:

$$\text{Inf}_{\text{Adam}}(z, z') = \sum_{i=1}^T \bar{\eta}_i \cos\left(\nabla_\theta \ell(z'; \theta_i), \Gamma(z, \theta_i)\right), \tag{1}$$

where $\bar{\eta}_i$ is the average learning rate in epoch $i$ (out of $T$ total epochs) and $\theta_i$ is the checkpoint after epoch $i$. $\Gamma$ requires the moment statistics $(\boldsymbol{m}, \boldsymbol{v})$, which depend on past gradients, with details provided in Appendix B.

**Why gradients, not embeddings?** Empirically (Fig. 1b and 1c), we observe two phenomena: (i) synthetic sets whose gradient distributions are closer to the validation distribution yield better downstream performance; (ii) the same trend does *not* hold in embedding space, where proximity often fails to predict gains. This explains why aesthetically "high-quality" samples may not train well: they align semantically but steer optimization in suboptimal directions. Let $\text{IF}(Q, A)$ denote the Adam-compatible influence score (Eq. 1) of the synthetic pair $(Q, A)$ with respect to the validation set. We further quantify this by correlating IF averages with held-out accuracy (Fig. 1a): IF exhibits a strong positive correlation, validating it as a reliable proxy for synthetic data quality.

**Takeaway.** Influence functions provide a principled, model-aware signal for judging the *training utility* of synthetic examples, turning data synthesis from heuristic rubric engineering into *model-centric* optimization. Building on this, we close the loop with an RL framework in which a *rubric generator* acts as the agent (replacing ad-hoc human design), explores the rubric space, and receives influence-based rewards that estimate each synthesized sample's contribution to downstream improvement. This alignment of generation with measured model impact yields more useful data with less domain-specific handcrafting, better sample efficiency, and a clear path to automatic refinement across tasks and domains.

## 2.3 INFLUENCE-GUIDED REINFORCEMENT LEARNING

We train the *rubric generator* as a policy that explores the rubric space and receives *influence-based* feedback from the target model. Formally, given a seed datum $S$, the policy LLM $\pi_\theta$ produces a rubric $B \sim \pi_\theta(\cdot | S)$; a teacher model then conditions on $(S, B)$ to synthesize a pair $(Q, A)$; the target model supplies a scalar reward that scores the *training utility* of $(Q, A)$. A trajectory is $\tau = \{S, B, (Q, A)\}$, and the objective is to maximize $\mathbb{E}_{\tau \sim \pi_\theta}[R(\tau)]$.

**Verifiable, influence-based rewards.** We combine automatic validity checks with an optimizer-aware influence estimate. Let $\text{Valid}(Q, A) \in \{0, 1\}$ be a conjunction of lightweight verifiers (for-

matting, non-triviality, safety). The reward is:

$$R(\tau) = \text{Valid}(Q, A) \cdot \text{IF}(Q, A) - \lambda(1 - \text{Valid}(Q, A)), \tag{2}$$

where $\lambda > 0$ penalizes invalid generations. This yields a *verifiable* signal that is aligned with downstream improvement while discouraging degenerate outputs.

**Policy optimization.** We use a clipped policy-gradient objective (GRPO/PPO-style) with group-relative baselines for variance reduction. For each seed $S$ we sample $G$ rubrics, producing trajectories $\{\tau_i\}_{i=1}^G$. Denote by $\tau_{i,(t)}$ the $t$-th token and by $\tau_{i,<t}$ its prefix. The objective is

$$J(\theta) = \frac{1}{G} \sum_{i=1}^G \frac{1}{|\tau_i|} \sum_{t=1}^{|\tau_i|} \min\Big[r_{i,t}(\theta)\,\hat{A}_{i,t},\ \text{clip}\big(r_{i,t}(\theta),\, 1-\epsilon,\, 1+\epsilon\big)\hat{A}_{i,t}\Big] - \beta\,\text{KL}\big(\pi_\theta \,\|\, \pi_{\text{ref}}\big), \tag{3}$$

with importance ratio $r_{i,t}(\theta) = \pi_\theta\big(\tau_{i,(t)} \mid \tau_{i,<t}\big) / \pi_{\text{old}}\big(\tau_{i,(t)} \mid \tau_{i,<t}\big)$, and a group-normalized advantage $\hat{A}_{i,t} = \big(R(\tau_i) - \frac{1}{G}\sum_{j=1}^G R(\tau_j)\big) / \sqrt{\frac{1}{G}\sum_{j=1}^G \big(R(\tau_j) - \frac{1}{G}\sum_{k=1}^G R(\tau_k)\big)^2 + \delta}$, where $\epsilon$ is the clipping parameter, $\beta$ controls KL trust region against a reference policy $\pi_{\text{ref}}$ (e.g., the SFT model), and $\delta > 0$ stabilizes normalization. We add entropy regularization to encourage exploration.

This RL loop closes the synthesis–training feedback cycle: the policy learns rubrics that systematically *maximize measured training impact*, yielding synthetic data that is not only plausible by heuristic criteria but empirically *helpful for learning*.

## 3 EXPERIMENT

### 3.1 EXPERIMENTAL SETUP

**Models** The *teacher LLM* (also denoted as generator), unless otherwise specified, is *Qwen3-235B-Instruct* (Yang et al., 2025), which generates synthetic supervised fine-tuning (SFT) data; in ablation studies, we substitute it with *GPT-4.1* (Achiam et al., 2023) and *Gemini-2.5-Pro* (Comanici et al., 2025) to evaluate the sensitivity of our pipeline to the teacher model choice. The *target LLM* is by default *Qwen3-8B-Base*, trained on the synthesized data; we further examine in two settings: (i) replacing the target with *Llama3-8B-Base* (Dubey et al., 2024) and (ii) Qwen variants—*Qwen3-4B-Base*, *Qwen3-8B-Base*, and *Qwen3-14B-Base*—using the *same* learned prompter. Finally, the *prompter base model* is initialized from *Qwen3-8B-Instruct*, which serves as the base model for rubric optimization throughout training.

**Training Dataset** We apply our SFT data–synthesis framework to two *Humanities and Social Sciences* and *Medical and Health*. For HSS, we curate seed documents from books: specifically, we select 223 humanities and social sciences books with high public ratings on *Goodreads*, with the domain distribution shown in Fig. 3 (left). Texts are segmented by table-of-contents chapters, and we remove boilerplate, front/back matter, and other non-textual or noisy content. For the medical domain, we draw seed documents from the pretraining corpora released with the Meditron large language model (Chen et al., 2023) and from a subset of open-access PubMed abstracts; the distribution is shown in Fig. 3 (right). Because rubric generation requires feeding each seed document to the prompter

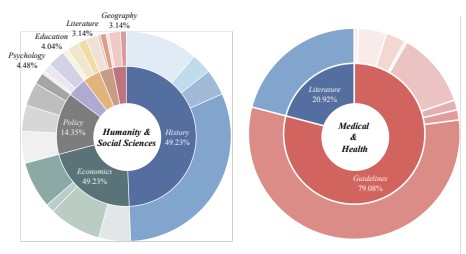

Figure 3: Domain distribution of seed documents used to synthesize question-answering pairs: *Humanities and Social Sciences* and *Medical and Health*.

model and is constrained by the model's input window, we retain only documents shorter than 8k tokens. After filtering, the HSS and medical corpora contain 18,665 and 26,435 seed documents, respectively. Further details of the raw sources are provided in Appendix D.3.

**Validation Set Construction** When constructing the validation set, we follow two principles: (i) there is no overlap or information leakage with the SFT training data, and (ii) the validation data

Table 1: Results across two domains. All metrics are accuracy (higher is better). Across both HSS and Medical domains, our method consistently upgrades Qwen3-8B-Base beyond widely used open SFT corpora, often rivaling or surpassing Qwen3-8B-Instruct.

| Humanities and Social Sciences (HSS) | | | | | | |
|---|---|---|---|---|---|---|
| **Model** | **MMLU pro** | **Super GPQA** | **HLE** | **BBH** | **HellaSwag** | **DROP** |
| Qwen3-8B-Base | 22.83 | 20.77 | 5.70 | 35.62 | 35.71 | 45.73 |
| Qwen3-8B-Instruct | 49.87 | 23.44 | 4.66 | **85.21** | 78.71 | **80.73** |
| MAGACorpus | 39.90 | 21.18 | **8.29** | 58.21 | 72.88 | 62.53 |
| Bonito | 39.37 | 21.36 | 5.70 | 45.78 | **83.00** | 50.95 |
| Conder | 41.99 | 23.29 | 6.22 | 74.21 | 59.32 | 41.52 |
| Cosmospedia | 27.30 | 20.03 | 5.18 | 55.63 | 79.33 | 49.70 |
| Longwriter | 30.45 | 12.02 | 6.22 | 34.89 | 65.09 | 23.62 |
| Openhermes | 49.56 | 24.60 | 3.63 | 73.64 | 73.41 | 61.62 |
| Synthquestion | 51.71 | 23.99 | 5.18 | 74.24 | 80.20 | 61.88 |
| WebrPro | 50.39 | 22.85 | 7.77 | 68.32 | 77.60 | 58.77 |
| Wildchat | 52.76 | 22.11 | 6.74 | 76.32 | 73.41 | 54.31 |
| **Ours** | **56.96** | **26.07** | 7.85 | 75.65 | 78.08 | 65.42 |
| Medical and Health | | | | | | |
| **Model** | **MMLU pro** | **Super GPQA** | **HLE** | **Health Bench** | **PubMed** | **MedQA** |
| Qwen3-8B-Base | 45.97 | 28.06 | 10.36 | 70.06 | 65.90 | 51.45 |
| Qwen3-8B-Instruct | 60.39 | 37.16 | 7.21 | **87.70** | 65.70 | 57.09 |
| ChatDoctor | 16.48 | 21.22 | 6.31 | – | **85.40** | 21.91 |
| MedQA | 13.45 | 34.61 | 9.91 | 30.46 | 71.70 | 53.41 |
| Medical-o1 | **60.51** | 27.26 | 8.11 | 70.14 | 73.90 | **58.75** |
| Medical-R1-Distill | 57.33 | 35.28 | 4.95 | 80.38 | 73.40 | 57.81 |
| ReasonMed | 55.00 | 28.01 | **14.41** | 51.76 | 79.80 | 23.56 |
| **Ours** | 56.97 | **38.28** | 10.81 | 74.82 | 80.70 | **58.75** |

come from public benchmark splits that share the same distribution as the downstream evaluations. In the HSS domain, we use the dev split of MMLU-pro (History) as the validation set; in the Medical domain, we use the official MedQA test split as the validation set. To ensure that the datasets used for synthetic data evaluation do not leak information from out-of-domain benchmarks, we follow the official AllenAI Tülu decontamination toolkit, implemented on Elasticsearch, to conduct leakage checks; details are provided in Appendix D.1.

**Evaluation Benchmarks** We evaluate models on 12 benchmarks in two domains. Because the humanities and social sciences (HSS) lack dedicated evaluation suites, we extract HSS-related subsets from three comprehensive datasets: MMLU-pro (*History*) (Wang et al., 2024), SuperGPQA (*History*) (Du et al., 2025), and Humanity's Last Exam (Phan et al., 2025) (*Humanities / Social Science*). We include Big Bench Hard (Kazemi et al., 2025), HellaSwag (Zellers et al., 2019), and DROP to gauge performance on complex comprehension and commonsense reasoning. For the medical domain, we assess capabilities using MMLU-pro (*Health*), SuperGPQA (*Health*), Humanity's Last Exam (*Biology / Medicine*), HealthBench (*Consensus*) (Arora et al., 2025), PubMedQA (Jin et al., 2019), and MedQA (Koh & Liang, 2017). Our evaluation adopts a 5-shot for MMLU-pro and a one-shot for PubMedQA and MedQA, with answer Accuracy as the primary metric. For benchmarks HLE and HealthBench that require an LLM-as-Judge, we use GPT-4.1 as the judge.

**Baselines** In the *humanities and social sciences*, our baselines comprise Qwen3-8B-Instruct (without additional fine-tuning) and models fine-tuned on eight SFT datasets. The datasets span three categories: (i) human-authored—WildChat (Zhao et al., 2024) and OpenHermes (Teknium, 2023); (ii) semi-automatically synthesized—MGACorpus (Hao et al., 2025), WebR Pro (Jiang et al., 2025), Bonito (Nayak et al., 2024), and SynthQuestions (Zhu et al., 2025); and (iii) fully automated synthetic—Conder (Cao et al., 2025). In the *medical and health* domain, our baselines include Qwen3-8B-Instruct and models fine-tuned on five SFT datasets: the real-world clinical dialogue corpus ChatDoctor (Li et al., 2023), the multiple-choice dataset MedQA (Singhal et al., 2025),

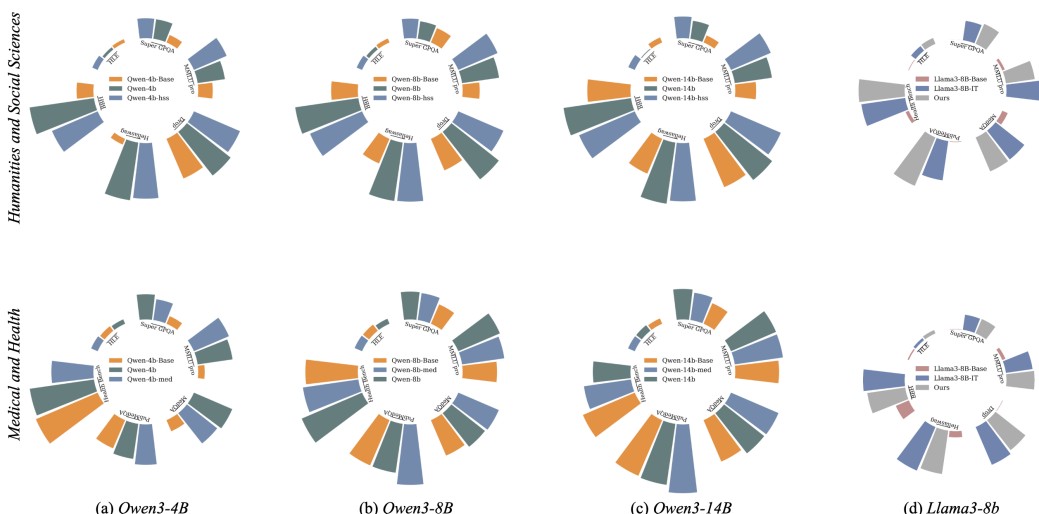

Figure 4: The performance of our framework across different target model scales and families.

| Dataset | Total samples | Token mean | Token std | MTLD | HDD |
|---|---|---|---|---|---|
| LongWriter | 6,000 | 1823.45 | 452.20 | 80.63 | 0.8954 |
| WildChat | 529,428 | 289.59 | 344.77 | 52.70 | 0.9188 |
| Condor | 20,000 | 428.79 | 35.70 | 101.48 | 0.8650 |
| Cosmopedia | 50,000 | 773.85 | 44.94 | 111.52 | 0.8843 |
| Openhermes 2.5 | 1,001,551 | 236.14 | 249.04 | 106.43 | 0.8887 |
| SynthQuestions | 2,500 | 634.60 | 17.88 | 137.02 | 0.8584 |
| OptimSyn (Ours) | 25,875 | 196.49 | 34.69 | 133.82 | 0.9241 |

Table 2: Statistics of different datasets (*Medical*).

and three LLM-distilled medical reasoning datasets—Medical-o1 (Chen et al., 2024), Medical-R1-Distill (Chen et al., 2024), and ReasonMed (Sun et al., 2025). We conduct a more detailed analysis of the data characteristics of the baseline datasets, such as data scale and diversity. For our proposed dataset and the previously used datasets, we report their statistical properties, including the number of samples, the mean and standard deviation of input/output lengths, as well as the lexical diversity of inputs and outputs (measured by MTLD and HDD), as shown in the Table 2. A detailed description of the data characteristics of the baseline dataset can be found in Appendix D.2.

**Implementation Details** For influence estimation, we first warm up the target model using 10% of the synthetic data generated by the initialized prompter and generator. The resulting model provides reference gradients for computing influence scores, which are subsequently incorporated into the RL stage. For reinforcement learning, we adopt GRPO (Guo et al., 2025) as the optimization algorithm, where the reward function integrates the influence score with a format-consistency component. Training is performed with a batch size of 256, learning rate of $1 \times 10^{-6}$, rollout temperature 1.5, rollout size $n = 5$, and one epoch of updates. $\lambda$ is set to 0.1. Processing approximately 20K samples takes about 10 hours. All experiments are conducted on $8 \times$H200 GPUs. Additional implementation details are provided in the Appendix D.4.

## 3.2 MAIN RESULTS

**Superiority over Human-Annotation and Synthetic-Data Baselines.** We present results for the two domains in Table 1. Our rubric-guided synthesis consistently enhances the target base model across both Humanities & Social Sciences and Medical domains, surpassing strong open SFT baselines and its instruct model. On HSS tasks, it rivals or even exceeds deliberate-reasoning teachers, achieving a +27.2% relative gain on HLE (0.0785 vs. 0.0570), thereby demonstrating that structured, group-aware data synthesis can effectively distill reasoning ability *without* test-time reasoning. In

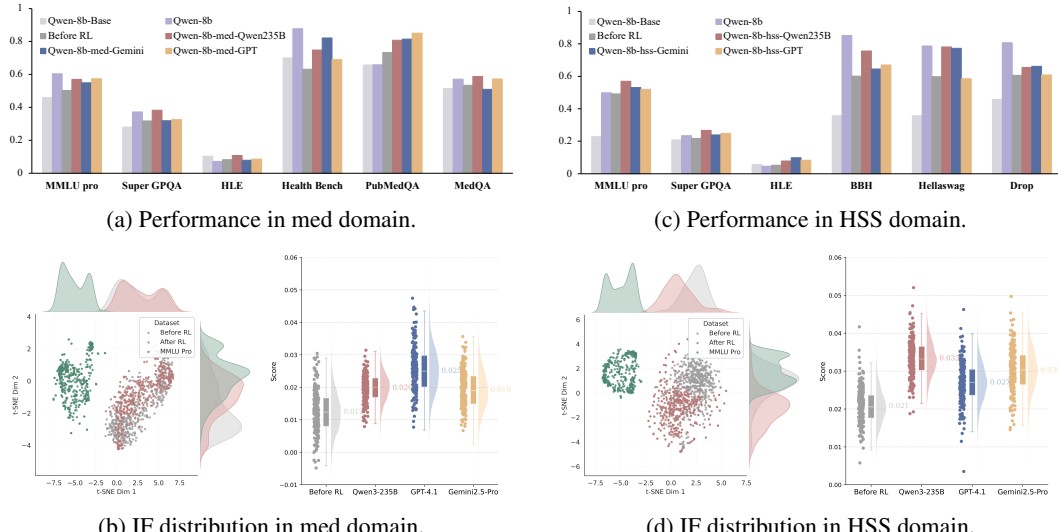

(a) Performance in med domain. (c) Performance in HSS domain.

(b) IF distribution in med domain. (d) IF distribution in HSS domain.

Figure 5: The performance of our framework across different generators.

the medical domain, our method narrows the gap on MMLU_PRO and SUPER_GPQA while decisively outperforming baselines on HLE and HEALTHBENCH. The most significant improvements appear on tasks aligned with domain rubrics, underscoring the strength of our prompter-centric RL framework in producing high-quality, domain-grounded supervision. Overall, the approach consistently upgrades the same 8B backbone beyond widely used SFT corpora, often matching or surpassing inference-time "thinking" models on reasoning-centric metrics.

**Generalization across Target Model Scales and Families.** To test whether our training recipe is target-agnostic, we replace the backbone with *Qwen3-{4B, 8B, 14B}* and *Llama3-8B* and evaluate on two domain suites—*Humanities & Social Sciences* and *Medical & Health*, with results shown in Figure 4. Across all four backbones, the plots exhibit consistent improvements for **Ours** than for the *Base* and *Instruct* variants on most tasks within both domain suites. The gains persist when moving from small to larger models (4B → 14B), while remaining substantial at the smallest scale, suggesting that our method does not rely on model capacity to be effective. Moreover, the improvements transfer from the Qwen3 family to Llama3 with comparable margins, despite architectural and pretraining differences. Taken together, these results demonstrate that the proposed method confers *family- and scale-robust* benefits, supporting the claim that our method yields broadly applicable performance gains for diverse target LLMs.

**Robustness to the Choice of Generator.** We ablate the synthetic–data *generator* by swapping Qwen3–235B, GPT-4.1, and Gemini-2.5-Pro while keeping the target model fixed. The results are provided in Figure 5a and 5c. Across both domains—*Medical & Health* and *Humanities & Social Sciences*—models trained with our method consistently outperform the base model on nearly all benchmarks, indicating that the gains are not tied to any single generator.

## 3.3 ANALYSIS

**Consistent Enhancement of the Influence Distribution in Synthetic Data.** We further visualize the distribution of influence scores produced by our method, as shown in Figure 5d and Figure 5b. The left panel shows that our approach aligns the distribution of synthetic data influence scores more closely with that of the validation data. The right panel demonstrates that, even when substituting different generators, our method consistently shifts the distribution toward higher average influence scores. Taken together, these results indicate that the proposed reward signal effectively selects and amplifies high-utility synthetic examples in a *generator-agnostic* fashion.

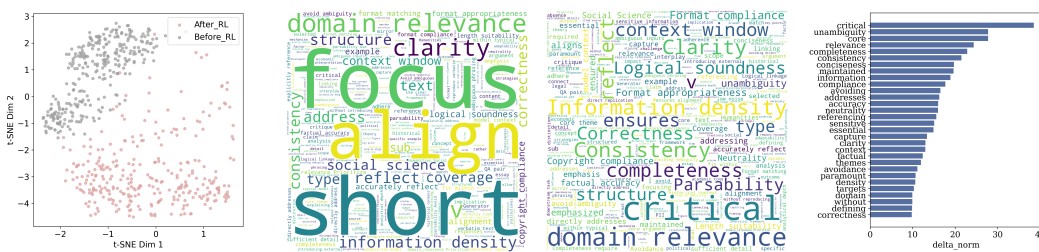

(a) Distributions of the embeddings of rubrics.    (b) Word cloud of rubrics without our method.    (c) Word cloud of rubrics with our method.    (d) New high-frequency terms in Figure 6c.

Figure 6: Effects of the Proposed Method on Rubrics

**Effects of the Proposed Method on Rubrics.**    We examine how the prompter-centric RL reshapes the learned rubrics. Figure 6a visualizes t-SNE embeddings of rubric texts in the generator space before and after RL: post-RL rubrics occupy a broader, more structured region, whereas pre-RL rubrics cluster in a narrow band, indicating increased coverage and diversity aligned with the seed data distribution. From our case analyses, we observe that the post-RL rubrics are more tightly grounded in the source documents, which in turn explains their broader coverage and clearer structure in the embedding space. We hypothesize that such more detailed rubrics better steer the generator to synthesize data that conforms to the specification and task requirements. The word clouds in Figure 6b–6c reveal a qualitative shift from generic imperatives (e.g., *focus*, *align*, *short*) to domain- and quality-oriented attributes (e.g., *critical*, *clarity*, *completeness*, *parsability*, *information density*, *logical soundness*). Figure 6d further quantifies these changes by highlighting newly introduced high-frequency terms and their normalized frequency deltas, confirming that our method promotes specific, actionable criteria rather than vague instructions. Detailed case studies of the rubrics and question–answer pairs are provided in Appendix E.8 and E.9.

**Predictive Relationship Between Influence Scores and Downstream Accuracy.**    We investigate whether per-example Influence Scores (IF), computed on the *synthetic* training pool, are predictive of downstream supervised fine-tuning (SFT) performance. Specifically, we repeatedly sample random subsets of size 2K or 4K from the generated synthetic dataset, compute the aggregate IF of each subset, and fine-tune the model exclusively on that subset. Figure 7 shows that higher-IF subsets consistently yield higher test accuracy under both training budgets. Quadratic regressions (dashed curves) summarize the trend with strong goodness of fit ($R^2 = 0.57$ for 2K and $R^2 = 0.54$ for 4K), revealing a clear positive correlation with mild saturation at the high-IF end. These results demonstrate that IF serves as a reliable proxy for the utility of synthetic examples, and that prioritizing higher-IF data can improve downstream accuracy even with limited training budgets.

**Effects of the Rubrics on Synthetic Data.**    We ablate the number of rubric rollouts per seed, $G$, to assess how exploration affects synthetic data quality. Figure 8 shows RL training curves for $G \in \{5, 10, 15\}$ larger $G$ reaches higher reward and exhibits reduced variance, suggesting that broader rubric exploration stabilizes IF-driven optimization. Figure 9 reports downstream accuracy on multiple benchmarks: increasing $N$ consistently improves performance. Together, these results support that diverse, targeted rubrics—enabled by higher rollout counts—both *stabilize* training and *amplify* the utility of synthetic data.

## 4    RELATED WORKS

As LLMs improve, synthetic data is increasingly used to offset annotation costs for instruction tuning. A first line of work converts readily available document corpora into SFT-style dialogue data: WebR (Jiang et al., 2025) casts webpages as dual-view reconstruction (web-as-instruction / web-as-response with evidence-constrained refinement), enabling seedless, label-free instruction-tuning corpora; MAmmoTH2 (Yue et al., 2024) scales via a web-recall→IR-extraction→decontamination→refinement pipeline; Bonito maps unlabeled text plus task attributes to instruction–response via meta-template conditional generators, supporting

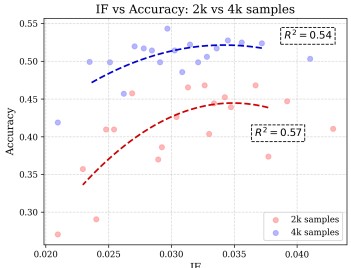

Figure 7: Influence score vs. accuracy.

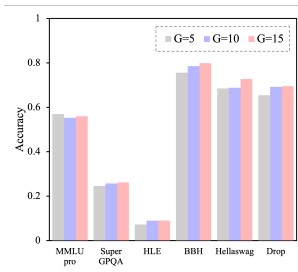

Figure 8: Accuracy across different group sizes.

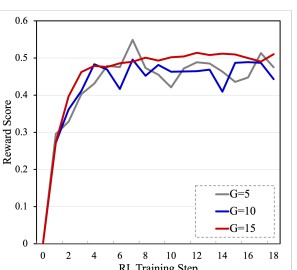

Figure 9: Training curves across different group sizes.

zero-shot cross-task/domain adaptation. A complementary line focuses on iteratively improving the synthesized data: Evol-Instruct (Xu et al., 2024) evolves seed instructions to increase difficulty and produce matched responses; Condor (Cao et al., 2025) organizes a World Knowledge Tree and applies self-reflection refinement (strengths/weaknesses/suggestions) in a two-stage knowledge-driven, self-reflective pipeline; (Li et al., 2024a) employs failure-inducing exploration by feeding student-model failures back to the teacher to synthesize targeted hard examples; and Montessori-Instruct (Li et al., 2024b) contrasts beneficial vs. non-beneficial samples and uses Direct Preference Optimization to bias the teacher toward the most instructive data.

Compared to prior work, our approach (i) replaces heuristic filters with a training-signal–aligned objective that directly optimizes downstream improvement; (ii) learns rubrics from model feedback rather than expert priors, enabling transfer across domains without bespoke rule engineering; and (iii) leverages influence-based analysis to reveal the mismatch between embedding similarity and training influence, clarifying failure modes of earlier pipelines and guiding principled corrections. By elevating the role of the target model's own training signal in the synthesis loop, our approach provides a principled route to scalable SFT in data-scarce, high-stakes domains. It turns rubric design from a brittle, expert-driven craft into a learnable component that aligns synthetic data generation with what *actually* improves the model.

## 5 CONCLUSION

We introduced a synthesis–training framework that replaces heuristic rubric design with *model–impact supervision*. At its core is an optimizer-aware, gradient-based influence estimator that quantifies each synthetic QA pair's contribution to the target model's objective, revealing a marked mismatch between embedding similarity and training impact and motivating gradient-aligned selection. Leveraging this signal, we cast rubric construction as a learnable policy within a prompter-centric RL loop, optimized with GRPO objectives under verifiable rewards that blend lightweight validity checks with influence scores. This closes the synthesis–training feedback cycle and grounds rubric learning in downstream impact rather than surface heuristics. Empirically, our approach consistently improves performance across domains, model families, and generators, rivaling or surpassing standard SFT baselines. Influence scores correlate strongly with realized accuracy under fixed budgets, validating them as a reliable proxy for synthetic data quality and highlighting the portability of our prompter-centric optimization framework.

## 6 LIMITATIONS

Our prompter is optimized via reinforcement learning using scalar rewards computed *after* a separate generator synthesizes QA pairs conditioned on the sampled rubrics. This introduces an indirect credit-assignment path from the target-model feedback to the prompter: the reward reflects both the generator's stochasticity and the downstream influence estimation, but gradients cannot be propagated through the generation step. As a result, policy-gradient updates (e.g., GRPO) can exhibit high variance and training instability, especially when the rollout group size is small.

## AKNOWLEDGEMENT

This work is supported by the National Key R&D Program of China (Grant No. 2024YFC3308304), the "Pioneer" and "Leading Goose" R&D Program of Zhejiang (Grant No. 2025C01128), the National Natural Science Foundation of China (Grant No. 62476241), and the Natural Science Foundation of Zhejiang Province, China (Grant No. LZ23F020008).

## ETHICS STATEMENT

All authors attest that they have read and will abide by the ICLR Code of Ethics. Our study develops an influence-aligned, prompter-centric framework for rubric-guided synthetic data generation and evaluates it on knowledge-intensive domains. Below we summarize ethical considerations specific to this work.

**Human subjects and IRB.** This work does not involve human participants, user studies, or the collection of personal data. No IRB approval was required.

**Data sources, licensing, and privacy.** Seed documents are drawn from publicly available sources in the Humanities Social Sciences and Medical domains; preprocessing removes boilerplate and noisy content, and the generator is instructed to paraphrase rather than reproduce lengthy excerpts verbatim. We release only prompts/rubrics and synthesized QA pairs, not the original documents. We screen out personally identifiable information (PII) and sensitive attributes to the best of our ability and respect the licenses and terms of use of all sources. We will honor takedown requests for inadvertently included material.

**Safety, misuse, and domain constraints.** Although our method targets data quality, synthetic outputs in medical or other high-stakes settings may be misinterpreted as professional advice. Our models and datasets are research artifacts and *not* intended for deployment in clinical, legal, financial, or safety-critical contexts. The rubric pipeline includes format/safety checks (e.g., refusal of unsafe instructions) and excludes harmful topics during synthesis. We recommend downstream users apply additional content filters and human review.

**Use of external APIs and compliance.** When large closed models are used as generators or judges, all interactions comply with provider terms of service; no user data or proprietary content is transmitted beyond the required prompts. We report which systems are used and how (teacher, judge) in the paper.

**Conflicts of interest and sponsorship.** The authors declare no conflicts of interest related to this submission. No external sponsor influenced the problem formulation, experiments, or reporting.

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

## A    THE USE OF LARGE LANGUAGE MODELS

In the preparation of this manuscript, GPT-5 was employed solely for language refinement purposes, including grammar checking and stylistic polishing. All conceptualization, methodological design, experiments, analyses, and interpretations were conducted by the authors. The LLM was not used to generate novel content, ideas, or results, but only to assist in improving clarity, readability, and consistency of expression. Specifically includes: 1 INTRODUCTION; 2 METHOD; 4 RELATED WORKS; 5 CONCLUSION.

## B    INFLUENCE ESTIMATION

Influence estimation Koh & Liang (2017); Pruthi et al. (2020) aim to quantify how an individual training example perturbs both the learned parameters and the loss at a target (validation/test) point. In the large–language–model (LLM) regime, computing exact influences is prohibitively expensive; consequently, prior work Pruthi et al. (2020); Xia et al. (2024) adopts a first-order approximation to the training dynamics to estimate the influence of a training datapoint on held-out performance.

**Per-step influence.**    Let $\theta^t$ denote the model parameters at step $t$ and $\ell(\cdot; \theta^t)$ the loss. A first–order Taylor expansion of the loss on a validation example $z'$ gives

$$\ell(z'; \theta^{t+1}) \approx \ell(z'; \theta^t) + \langle \nabla_\theta \ell(z'; \theta^t), \theta^{t+1} - \theta^t \rangle. \tag{4}$$

Assuming SGD with batch size 1 and learning rate $\eta_t$, if $z$ is used at step $t$ then $\theta^{t+1} - \theta^t = -\eta_t \nabla_\theta \ell(z; \theta^t)$, and equation 4 yields the per-step influence

$$\ell(z'; \theta^{t+1}) - \ell(z'; \theta^t) \approx -\eta_t \langle \nabla_\theta \ell(z; \theta^t), \nabla_\theta \ell(z'; \theta^t) \rangle. \tag{5}$$

(For mini-batches, replace $\nabla_\theta \ell(z; \theta^t)$ with the batch gradient.)

**Trajectory influence (SGD).**    Aggregating equation 5 over epochs results in a trajectory-level score:

$$\text{Inf}_{\text{SGD}}(z, z') = \sum_{i=1}^{N} \bar{\eta}_i \langle \nabla_\theta \ell(z'; \theta_i), \nabla_\theta \ell(z; \theta_i) \rangle, \tag{6}$$

where $\bar{\eta}_i$ is the average learning rate in epoch $i$ (out of $N$ total epochs) and $\theta_i$ is the checkpoint after epoch $i$.

**Extension to Adam.**    Instruction tuning typically uses Adam (Adam et al., 2014). Let the per-coordinate update direction be

$$\Gamma(z, \theta^t) \triangleq \frac{\hat{\boldsymbol{m}}^{t+1}}{\sqrt{\hat{\boldsymbol{v}}^{t+1}} + \epsilon}, \quad \theta^{t+1} - \theta^t = -\eta_t \Gamma(z, \theta^t), \tag{7}$$

with elementwise operations and

$$\boldsymbol{m}^{t+1} = \beta_1 \boldsymbol{m}^t + (1 - \beta_1) \nabla_\theta \ell(z; \theta^t),$$
$$\boldsymbol{v}^{t+1} = \beta_2 \boldsymbol{v}^t + (1 - \beta_2) \nabla_\theta \ell(z; \theta^t)^{\odot 2},$$
$$\hat{\boldsymbol{m}}^{t+1} = \boldsymbol{m}^{t+1} / (1 - \beta_1^{t+1}), \quad \hat{\boldsymbol{v}}^{t+1} = \boldsymbol{v}^{t+1} / (1 - \beta_2^{t+1}).$$

A first–order expansion analogous to equation 5 gives the stepwise effect $-\eta_t \langle \nabla_\theta \ell(z'; \theta^t), \Gamma(z, \theta^t) \rangle$. Because the magnitude of $\Gamma$ varies with the adaptive moments, we use a scale-robust cosine similarity when accumulating across epochs, and define

$$\text{Inf}_{\text{Adam}}(z, z') = \sum_{i=1}^{N} \bar{\eta}_i \cos\big(\nabla_\theta \ell(z'; \theta_i), \Gamma(z, \theta_i)\big). \tag{8}$$

Computing $\Gamma$ requires the moment statistics $(\boldsymbol{m}, \boldsymbol{v})$, which depend on past gradients; following our procedure in §3.1.3, we obtain them from a short warmup run before scoring.

## C  RELATED WORKS: SYNTHETIC DATA GENERATION

With the continued improvement of LLM capabilities, synthetic data has been widely used to alleviate the high annotation costs in instruction tuning and dialogue scenarios, many studies convert readily available document corpora into SFT-style dialogue samples using LLMs. WebR (Jiang et al., 2025) treats web pages as a dual-view reconstruction target: first, page content is converted into structured instructions and corresponding responses; second, the page is regarded as the response to infer latent instructions, after which the initial responses are refined under evidence constraints. This enables the construction of high-quality instruction-tuning corpora without human intervention or seed data. MAmmoTH2 (Yue et al., 2024) proposes a scalable pipeline comprising web recall, extraction of instruction–response pairs, decontamination, and refinement. Bonito (Nayak et al., 2024) maps unlabeled text and task attributes into instruction–response samples via meta-template–driven conditional task generators, turning domain documents into trainable data and enabling zero-shot cross-task and cross-domain adaptation. A complementary line of work focuses on iteratively improving the synthesized data. Evol-Instruct (Xu et al., 2024) evolves seed instructions in depth and breadth to progressively increase task difficulty and generate matched responses, illustrating the potential of self-evolving synthesis. Condor (Cao et al., 2025) constructs a World Knowledge Tree to ensure topic coverage and difficulty stratification and introduces a self-reflection refinement mechanism that conducts structured self-evaluation of candidate answers along strengths, weaknesses, and suggestions before rewriting, yielding a two-stage knowledge-driven and self-reflective synthesis framework. Li et al. (2024a) adopts failure-inducing exploration by feeding cases that the student model still fails on back to the teacher to synthesize targeted hard examples. Montessori-Instruct (Li et al., 2024b) forms pairs of beneficial and non-beneficial data and applies Direct Preference Optimization to train the teacher's preference toward examples that are most instructive for downstream performance.

Current data selection strategies can be broadly divided into two categories:(1) Static criteria (human-authored rubrics / LLM-as-Judge): Emphasize interpretability and control. Representative approaches include: QuRating (Wettig et al., 2024), which uses an LLM to score texts along dimensions such as quality, expertise, factuality/knowledge, and educational value to guide sample selection; DSIR (Xie et al., 2023), which applies importance resampling to large-scale pretraining subset selection; and FineWeb (Penedo et al., 2024), which performs large-scale deduplication and filtering and uses distributional visualizations and ablations to validate how data quality affects performance, thereby providing a tooling stack for curating high-quality general-purpose corpora. (2) Dynamic feedback (model-/task-driven): Directly selects samples using signals for what is most instructive for the model. Representative approaches include: LESS (Xia et al., 2024), which selects instruction samples via influence estimates and low-rank gradient similarity; PreSelect (Shum et al., 2025), which uses predictability as a lightweight scoring function; PDS (Gu et al., 2024), which formulates data selection as an optimal-control problem to characterize selection–training dynamics; and MATES (Yu et al., 2024), which learns a data-influence model that adapts over training to match the most effective samples for the current stage.

## D  EXPERIMENT DETAILS

### D.1  DECONTAMINATION PROTOCOL.

We follow the official AllenAI Tülu decontamination toolkit, implemented on Elasticsearch, and use three complementary matching strategies to mitigate train–test leakage. In text mode (without n-grams), we issue strict match-phrase queries over the validation index (used for influence-function computation) and flag any example that contains an identical contiguous span to the evaluation prompt, thus capturing verbatim duplicates. In n-gram mode, we tokenize each evaluation instance, construct 5-grams, retrieve validation documents containing these 5-grams, and compute an overlap score as the fraction of query tokens covered by matched n-grams; validation examples whose maximum overlap with any test item exceeds a threshold are removed, which detects high-lexical-overlap near duplicates. In vector mode, we encode both the validation and evaluation texts with a pre-trained embedding model (NV-Embed-v2), perform kNN search over the stored embeddings using Elasticsearch's vector index, and treat the highest similarity to any test item as a semantic contamination score, removing validation examples above a similarity threshold; this step identifies

| Method | MMLU-pro | Super GPQA | HLE | PubMed | MedQA | Health Bench |
|---|---|---|---|---|---|---|
| Text-match | 0.000 | 0.000 | 0.000 | 0.000 | 0.000 | 0.000 |
| 5-gram (0.3) | 0.002 | 0.004 | 0.000 | 0.005 | 0.009 | 0.001 |
| Embedding (0.85) | 0.000 | 0.001 | 0.000 | 0.001 | 0.009 | 0.002 |

Table 3: Estimated contamination scores on medical and health benchmarks under different decontamination modes.

| Method | MMLU-pro | Super GPQA | HLE | BBH | HellaSwag | DROP |
|---|---|---|---|---|---|---|
| Text-match | 0.000 | 0.000 | 0.000 | 0.000 | 0.000 | 0.000 |
| 5-gram (0.3) | 0.006 | 0.004 | 0.001 | 0.000 | 0.000 | 0.000 |
| Embedding (0.85) | 0.003 | 0.001 | 0.000 | 0.000 | 0.000 | 0.000 |

Table 4: Estimated contamination scores on general reasoning benchmarks under different decontamination modes.

paraphrased or semantically equivalent variants that purely lexical methods may miss. The detailed results are shown in Table 3, Table 4.

## D.2 BASELINE ANALYSIS

We conduct a more detailed analysis of the data characteristics of the baseline datasets, such as data scale and diversity. For our proposed dataset and the previously used datasets, we report their statistical properties, including the number of samples, the mean and standard deviation of input/output lengths, as well as the lexical diversity of inputs and outputs (measured by MTLD and HDD), as shown in the table below. As can be seen, existing SFT datasets already cover a very wide range in terms of these statistics. Compared with other datasets, the size, average length, and diversity metrics of OptimSyn all lie within this existing range. In other words, OptimSyn is better viewed as a more "efficient" data mixture obtained by re-organizing and filtering samples within the existing statistical range, rather than simply relying on "more data" or "longer texts". More importantly, when we relate these statistics to the downstream evaluation results (Table 2), we do not observe a monotonic trend that "larger scale / longer length leads to better performance": under the same target model and training configuration, some datasets with larger size or longer outputs do not achieve the best performance, whereas several moderately sized but higher-quality corpora yield more stable results instead. This suggests that basic corpus statistics alone are insufficient to explain the performance gains brought by our method.

## D.3 SOURCE OF TRAINING DATASET

Our seed documents for synthesizing data in the humanities and social sciences were collected from books. The list of book titles is as follows:

*A Black Jurist in a Slave Society Antonio Pereira Rebouças and the Trials of Brazilian Citizenship; A History of Intellectual Property in 50 Objects; A History of Modern Africa 1800 to the Present Wiley Blackwell Concise History of the Modern World; A Taste of Irrationality Sample Chapters From Predictably Irrational and Upside of Irrationality; A Case for Sometimes Tubefeeding Patients in Persistent Vegetative State; Adult Attachment A Concise Introduction to Theory and Research; Advanced Programming in the UNIX Environment 3rd Edition; Advanced Signal Integrity for Highspeed Digital Designs; Advocating for Self Womens Decisions Concerning Contraception; African History A Very Short Introduction; Algorithm Design Monograph; American Criminal Justice Policy An Evaluation Approach to Increasing Accountability and Effectiveness; American Grace How Religion Divides and Unites Us; An Analysis of Ernest Gellners Nations and Nationalism; An Analysis of Marcel Mauss The Gift The Form and Reason for Exchange in Archaic Societies; An Analysis of Robert A Dahls Who Governs Democracy and Power in an American City; An Analysis of Robert E Lucas Jrs Why Doesnt Capital Flow from Rich to Poor Countries; An Organizers Tale Speeches; Assessments in Forensic Practice A Handbook; Bargaining with the*

*Devil When to Negotiate When to Fight; Basic Political Writings; Bertrand Russell and the Nature of Propositions A History and Defence of the Multiple Relation Theory of Judgement; Beyond Reason Using Emotions as You Negotiate; Blackness in Britain; Challenging Behaviour in Dementia A Personcentred Approach; Changing Minds The Art and Science of Changing Our Own and Other Peoples Minds; Child Soldiers A Reference Handbook; Children and Play Understanding Childrens Worlds; Cite Them Right The Essential Referencing Guide 12th Edition; Cognitive Behavioral Therapy for PTSD A Case Formulation Approach; Cognitive Behavioral Treatment for Generalized Anxiety Disorder From Science to Practice; Collaborative Intelligence Using Teams to Solve Hard Problems; Collective Action and Exchange A Gametheoretic Approach to Contemporary Political Economy; Companion Encyclopedia of Medicine in the Twentieth Century; Computer Architecture A Quantitative Approach; Concise Guide to APA Style The Official APA Style Guide for Students; Confronting The Internets Dark Side Moral And Social Responsibility On The Free Highway; Contemporary Debates in Cognitive Science Contemporary Debates in Philosophy; Curing Their Ills Colonial Power and African Illness; Dependency and Development in Latin America; Descriptive Physical Oceanography Sixth Edition An Introduction; Development Microeconomics; Dignity and Daily Bread New Forms of Economic Organizing among Poor Women in the Third World and the First; Divine Hiddenness and Human Reason; Economics and Policy Issues in Climate Change; Economics of Regulation and Antitrust Fifth Edition The MIT Press; Effective Treatments for PTSD Third Edition Practice Guidelines from the International Society for Traumatic Stress Studies; European Legal History A Cultural and Political Perspective; Euthanasia Ethics and the Law from Conflict to Compromise; Exchange Rate Regimes Fixed Flexible or Something in Between; Experiences of Depression A Study in Phenomenology; Factfulness Ten Reasons Were Wrong About the World and Why Things Are Better Than You Think; Fatal Invention How Science Politics and Big Business Recreate Race in the Twentyfirst Century; Feeding the World An Economic History of Agriculture 1800–2000; Feminism Literature and Rape Narratives Violence and Violation; Feminism Unmodified Discourses on Life and Law; Feminist Judgments From Theory to Practice; Fiasco The American Military Adventure in Iraq 2003 to 2005; Formulation in Psychology and Psychotherapy Making Sense of Peoples Problems; Free Innovation; From the Great Recession to Labour Market Recovery Issues Evidence and Policy Options; Fundamentals of Developmental Psychology; Geographies of Development An Introduction to Development Studies; Getting It Wrong How Canadians Forgot Their Past and Imperilled Confederation; HBRs 10 Must Reads on Managing Strategy; Hackers Delight; Helping People Win at Work A Business Philosophy Called Dont Mark by Paper Help Me Get an A; Her Place at the Table A Womans Guide to Negotiating Five Key Challenges to Leadership Success; Hindu Worldviews Theories of Self Ritual and Reality; How To Do Your Research Project 3rd Edition; In Command of History Churchill Fighting and Writing the Second World War; Infertility and Patriarchy The Cultural Politics of Gender and Family Life in Egypt; Influence New and Expanded The Psychology of Persuasion; Innovation and Entrepreneurship Practice and Principles; Institutions Institutional Change and Economic Performance; Introduction to Econometrics Global Edition; Introduction to Meta-Analysis; Introductory Econometrics A Modern Approach MindTap Course List; Islam and Global Dialogue Religious Pluralism and the Pursuit of Peace; John P Kotter on What Leaders Really Do; Judgment How Winning Leaders Make Great Calls; Just Health Meeting Health Needs Fairly; Key Concepts in Geography; Law Legitimacy and the Rationing of Health Care A Contextual and Comparative Perspective; Leadership on the Line Staying Alive Through the Dangers of Leading; Lean In Women Work and the Will to Lead; Leviathan and the Air Pump Hobbes Boyle and the Experimental Life; Linear Algebra Done Right; Macleods Clinical Examination Ebook; Making Babies Is There a Right to Have Children; Making Democracy Work Civic Traditions in Modern Italy; Manias Panics and Crashes A History of Financial Crises; Marshall and the Marshallian Heritage Essays in Honour of Tiziano Raffaelli; Mastering Leadership A Vital Resource for Health Care Organizations; Math with Bad Drawings Illuminating the Ideas That Shape Our Reality; Microeconomics 9th Global Edition; Modeling Foundations of Economic Property Rights Theory An Axiomatic Analysis of Economic Agreements Studies in Economic Theory 23; Modern Classics Making of the English Working Class Penguin Modern Classics; Modern Environmentalism An Introduction; Money Well Spent A Strategic Plan for Smart Philanthropy; Mostly Harmless Econometrics; National Insecurity American Leadership in an Age of Fear; New Museum Theory and Practice An Introduction; Notes on the Theory of Choice; Oil Palm A Global History; One Economics Many Recipes Globalization Institutions and Economic Growth; Operating Systems In Depth Design and Programming; Ordinary and Partial Differential Equations With Special Functions Fourier Series and Boundary Value Problems; Organizational Trust A Reader; Paul Samuelson on the History of Economic Analysis Se-*

*lected Essays; Personality and Intelligence at Work Exploring and Explaining Individual Differences at Work; Philosophical Issues in Psychiatry Explanation Phenomenology and Nosology; Philosophy of Psychology Contemporary Readings; Presidential Leadership and the Creation of the American Era; Psychiatry as Cognitive Neuroscience Philosophical Perspectives; Psychological Assessment and Therapy with Older Adults; Psychology The Science of Mind Behaviour; Publication Manual of the American Psychological Association The Official Guide to APA Style; Raising Cando Kids Giving Children the Tools to Thrive in a Fastchanging World; Randomized Algorithms; Real Estate Development Principles and Process; Reconstructing Educational Psychology; Religion and Politics in the United States; Religion as Make Believe A Theory of Belief Imagination and Group Identity; Religion Violence Memory and Place; Research Methods and Statistics in Psychology 7th Edition; Rethinking Informed Consent in Bioethics; Reveille for Radicals; Rewriting the Soul Multiple Personality and the Sciences of Memory; Risk Ambiguity and Decision; Rule of Experts Egypt Technopolitics Modernity; Science and the Practice of Medicine in the Nineteenth Century; Seeing Like a State How Certain Schemes to Improve the Human Condition Have Failed; Selfcare Science Nursing Theory and Evidencebased Practice; Senior Leadership Teams What It Takes to Make Them Great; Slave Country American Expansion and the Origins of the Deep South; Small Differences That Matter Labor Markets and Income Maintenance in Canada and the United States; Small States in World Markets Industrial Policy in Europe; Smart Choices A Practical Guide to Making Better Life Decisions; Social Research; State Responsibility The General Part; Strategic Leadership and Management in Nonprofit Organizations Theory and Practice; Strategic Management for Nonprofit Organizations Theory and Cases; Strategy Safari A Guided Tour Through the Wilds of Strategic Management; Studies in Tudor and Stuart Politics and Government Papers and Reviews 1946–1972 Vol 1; Supervision in the Helping Professions Supervision in Context; The 5 Elements of Effective Thinking; The American Political Economy Macroeconomics and Electoral Politics; The Cambridge Companion to Shakespeare Cambridge Companions to Literature; The Cambridge Companion to Victorian and Edwardian Theatre Cambridge Companions to Literature; The Cambridge Handbook of Personality Psychology; The Economy of the Word Language History and Economics; The Far Enemy Why Jihad Went Global Second Edition; The Greatest Benefit to Mankind A Medical History of Humanity from Antiquity to the Present; The IBS Elimination Diet and Cookbook The Low FODMAP Plan for Eating Well and Feeling Great; The Little Black Book of Neuropsychology A Syndrome Based Approach; The Lucifer Effect Understanding How Good People Turn Evil; The Moral Economy of the Peasant Rebellion and Subsistence in Southeast Asia; The Mystery of Capital Why Capitalism Triumphs in the West and Fails Everywhere Else; The Politics of Life Itself Biomedicine Power and Subjectivity in the Twenty First Century; The Postcolonial Studies Reader; The Probabilistic Method; The Protestant Ethic and the Spirit of Capitalism; The Quest Energy Security and the Remaking of the Modern World; The Routledge Companion to Islamic Philosophy Routledge Philosophy Companions; The Routledge Handbook of Memory Activism; The SAGE Handbook of Social Geographies; The Social History of English Seamen 1485–1649; The United Nations and Changing World Politics Revised and Updated with a New Introduction; The Western Medical Tradition 1800–2000; The Wiley Blackwell Handbook of Individual Differences; The Architecture of the Mind Massive Modularity and the Flexibility of Thought; The Balanced Scorecard Translating Strategy into Action; The Blank Slate The Modern Denial of Human Nature; The Emergence of Industrial America Strategic Factors in American Economic Growth since 1870; The Female Frontier A Comparative View of Women on the Prairie and the Plains; The Filter Bubble What the Internet is Hiding from You; The Fortune at the Bottom of the Pyramid Eradicating Poverty Through Profits; The Global Interior Mineral Frontiers and American Power; The Missing Peace The Inside Story of the Fight for Middle East Peace; The Political Determinants of Health; The Prize The Epic Quest for Oil Money Power; The Psychological Complex Social Regulation and the Psychology of the Individual; The Slave Ship A Human History; To Vote or Not to Vote The Merits and Limits of Rational Choice Theory; Treatment of Obsessive Compulsive Disorder; Unified Growth Theory; Varieties of Capitalism The Institutional Foundations of Comparative Advantage; West from Appomattox The Reconstruction of America After the Civil War; Why David Sometimes Wins Leadership Organization and Strategy in the California Farm Worker Movement; Why Nations Fail The Origins of Power Prosperity and Poverty; Why We Disagree About Climate Change Understanding Controversy Inaction and Opportunity; Wind Energy Renewable Energy and the Environment; Working Bodies Interactive Service Employment and Workplace Identities; Yangzi Waters Transforming the Water Regime of the Jianghan Plain in Late Imperial China China Studies 44; The Courage to Create*

### D.4 Implementation Details

We conduct all experiments on $8 \times$H200 GPUs. The framework consists of three stages: warm-up, reinforcement learning (RL), and final SFT training.

**Warm-up.** At the beginning of training, the initialized prompter synthesizes a set of *rubrics*, which guide GENERATOR to produce an initial batch of SFT data. We then warm up the target model `qwen3-8b-base` by fine-tuning it with LoRA adapters and the Adam optimizer, using 10% of the synthetic data with batch size 16, learning rate $1!\times!10^{-5}$, and training for one epoch. The resulting model provides loss gradients as reference signals for influence score estimation. Each training example is assigned an *influence score*, which is normalized within the batch and used as the reward signal to update the prompter via reinforcement learning (RL).

**Reinforcement Learning.** During RL, a generator is deployed on $8 \times$H200 GPUs to synthesize QA pairs conditioned on seed documents and prompter rollouts. Each synthetic instance is scored by comparing the update direction (from Adam on the generated sample) with validation gradients of the target model. The similarity is then normalized via min–max scaling to $[0, 1]$, forming the primary reward signal. We adopt GRPO as the training algorithm, where the overall reward combines the influence score with format-consistency rewards. Training uses batch size 256, learning rate $1 \times 10^{-6}$, rollout temperature 1.5, rollout size $n = 5$, and one epoch of updates. Processing $\sim$20K samples requires roughly 10 hours.

**Final SFT.** After RL, the refined prompter generates an extended synthetic dataset, which is then used to conduct supervised fine-tuning (SFT) on `qwen3-8b-base` as the target model. This ensures that the model benefits both from high-quality rubric-guided data and from the RL-enhanced prompter optimization.

## E Experimental Results

### E.1 Baseline

Since Monressor-Instruct requires training the teacher model, we re-implement it by replacing its original teacher with Qwen-30B-A3B, so that all influence-based baselines and our method share the same strong teacher. Therefore, we argue that our work is not a simple application of existing influence-function tools to a new task, but rather a re-design of how influence-based signals are used under the realistic constraint of "synthetic data generation + a strong but frozen teacher": instead of "training the model", we shift to "training an evaluator, which in turn guides data synthesis". This usage pattern has been rarely discussed systematically in prior work and, in our opinion, constitutes the main conceptual novelty of our method. Under this unified setting, the results in Table 5 show a clear hierarchy among different ways of exploiting influence-based signals. Monressor-Instruct brings only marginal or even negative changes compared with the base model on several benchmarks (e.g., HLE and Health Bench), while the intermediate variant Monressor (inter 2) can leverage influence scores more effectively and yields consistent gains across almost all tasks. Our method further improves over these baselines on most general- and domain-specific benchmarks: it achieves the best performance on MMLU-pro, Super GPQA, HLE, Health Bench, and PubMed, and remains competitive on MedQA, slightly below Monressor (inter 2) but comparable to Monressor-Instruct. Taken together, these findings support our design choice of using influence-based signals to steer synthetic data construction via a learned evaluator, and demonstrate that this re-purposed pipeline can provide more robust and efficient improvements than directly optimizing the teacher model itself.

### E.2 Ablation on feedback mechanisms.

We conduct an ablation study that directly compares several representative feedback mechanisms under a unified setup, where we only vary the signal used to optimize the rubrics and select synthetic samples. Concretely, we consider (i) **QuRating** Wettig et al. (2024), which uses LLM-based pairwise quality scores along multiple dimensions (style, expertise, factuality, educational value); (ii) a **FineWeb-Edu**-style proxy Penedo et al. (2024), which measures the "educational / knowledge

| Method | MMLU-pro | Super GPQA | HLE | Health Bench | PubMed | MedQA |
|---|---|---|---|---|---|---|
| Base | 0.4597 | 0.2806 | 0.1036 | 0.7006 | 0.6590 | 0.5145 |
| Monressor-Instruct | 0.4672 | 0.2728 | 0.0946 | 0.6453 | 0.7027 | 0.5204 |
| Monressor (inter 2) | 0.5028 | 0.3105 | 0.1047 | 0.6893 | 0.7458 | 0.5469 |
| Ours | 0.5369 | 0.3549 | 0.1096 | 0.7062 | 0.7859 | 0.5206 |

Table 5: Performance comparison of different influence-based methods.

| Method | MMLU-pro | Super GPQA | HLE | Health Bench | PubMed | MedQA |
|---|---|---|---|---|---|---|
| Base-Model | 0.4597 | 0.2806 | 0.1036 | 0.7006 | 0.6590 | 0.5145 |
| QuRating | 0.4954 | 0.3032 | 0.1118 | 0.7106 | 0.7356 | 0.5534 |
| FineWeb-Edu | 0.4682 | 0.2892 | 0.1064 | 0.7340 | 0.7044 | 0.5628 |
| PRRC | 0.5317 | 0.3472 | 0.0914 | 0.7556 | 0.7568 | 0.5247 |
| Ours | 0.5697 | 0.3828 | 0.1081 | 0.7482 | 0.8070 | 0.5875 |

Table 6: Comparison of different reward / feedback mechanisms.

density" of each candidate sample by its similarity to the FineWeb-Edu distribution; (iii) **PRRC / Meta-rater** Zhuang et al. (2025), which aggregates multi-dimensional quality assessments (Professionalism, Readability, Reasoning, Cleanliness) into a single proxy-reward / proxy-validation score; and (iv) **Ours (Influence)**, which uses an influence-based signal aligned with the target validation loss. Results are shown in Table 6

On six medical and health benchmarks, all three proxy-based feedback mechanisms significantly improve over the base model, indicating that downstream reward or proxy-validation signals are effective. However, when averaging across all benchmarks, our influence-based feedback achieves the best overall performance. While proxy signals rely on heuristic properties of each sample (e.g., teacher scores, similarity to high-quality corpora, or performance on a small validation subset), our influence signal is derived from an optimization perspective and explicitly approximates how a small upweighting of a sample would change the target validation loss, thereby aligning data selection more tightly with the final training objective.

### E.3 ANALYSIS

**Impact of data scale.** We compare the effect of different amounts of synthetic data on model training. For the 8B base model, adding a small amount of synthetic data (5k–10k) already yields clear improvements over the base model, and performance continues to increase up to around 25k–30k examples on most benchmarks; beyond this point, enlarging the dataset to 40k brings diminishing or even slightly negative returns. In contrast, for the 14B base model, performance on MMLU-pro, Super GPQA, PubMed and MedQA improves more steadily as the synthetic data scale grows, with the best overall results obtained at 40k examples, although individual tasks such as Health Bench peak earlier. These trends suggest that the optimal data scale is both model-size- and task-dependent: smaller models quickly saturate and can even be hurt by excessive synthetic data, whereas larger models are better able to exploit larger-scale synthetic corpora.The specific results are shown in Figure 10.

### E.4 VALIDATION SET OF IF SCORE CONSTRUCTION AND ANALYSIS.

When constructing the validation set, we follow two principles: (i) no overlap or information leakage with the SFT training data, and (ii) the validation data should come from public benchmark splits that share the same distribution as the downstream evaluations. In the HSS domain, we use the dev split of MMLU-pro (History) as the validation set, while reporting test performance on the official test splits of MMLU-pro, SuperGPQA, and HLE. In the Medical domain, we use the official MedQA test split as the validation set for computing influence scores and guiding synthetic data selection, whereas evaluation is conducted on the Medical & Health suite including MMLU-pro (Health), SuperGPQA (Health), HLE, HealthBench, PubMedQA, and MedQA. This setup is con-

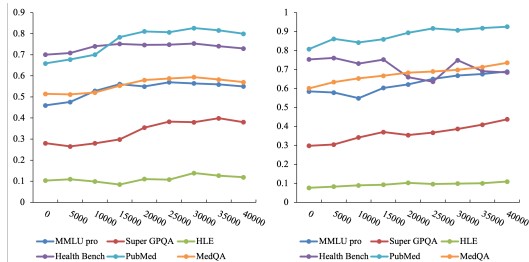

Figure 10: The impact of training data volume on model performance.

| Scale | MMLU-pro | Super GPQA | Health Bench | PubMed | MedQA | HLE | T-test |
|-------|----------|-----------|--------------|--------|-------|-----|--------|
| 8B-base | 0.4597 | 0.2806 | 0.7006 | 0.6590 | 0.5145 | 0.1036 | 0.0112* |
| 100 | 0.5236 | 0.3016 | 0.6759 | 0.6946 | 0.5141 | 0.0942 | 0.0046** |
| 300 | 0.5423 | 0.3096 | 0.6837 | 0.6480 | 0.4908 | 0.1006 | 0.0224* |
| 500 | 0.5318 | 0.3426 | 0.6995 | 0.7682 | 0.5738 | 0.0913 | 0.0023** |
| 800 | 0.5721 | 0.3739 | 0.7958 | 0.7835 | 0.5729 | 0.0982 | 0.9158 |
| 1000 | 0.5613 | 0.3663 | 0.7831 | 0.7961 | 0.6063 | 0.1057 | 0.7644 |
| 1273 | 0.5697 | 0.3828 | 0.7482 | 0.8070 | 0.5875 | 0.1081 | – |

Table 7: Effect of validation set size on downstream performance (8B model).

sistent with prior work such as LESS Xia et al. (2024), where an in-distribution public split is used purely as a validation set to compute gradients and influence scores, but is never included in SFT training, thereby preserving evaluation fairness while providing a representative task signal. We further analyze how the choice of validation set affects model performance in the Medical domain. Keeping the SFT data and training configuration fixed, we sample subsets of different sizes (100 / 300 / 500 / 800 / 1000 / full 1273 examples) from the MedQA test set as the validation set $D_{\text{val}}$ and use them to compute influence scores. As shown in Table 7, enlarging the validation set from 100 to 800 examples leads to steady improvements on downstream benchmarks, while the gains beyond 800 examples become marginal. Even very small validation sets (100 or 300 examples) already yield models that significantly outperform the 8B base model on average, but the performance variance across different random subsets is noticeably higher. To mitigate the instability of small validation sets, we explore two diversity-aware sampling strategies with a fixed validation size of 100: (i) *Embedding-Based*, which uses Qwen3-8B-Embedding to map candidate QA pairs into an embedding space and selects a subset with more uniform coverage in this space; and (ii) *EvalTree-Based*, which leverages the EvalTree capability decomposition to choose samples that cover a broad range of skills, especially those where the model is relatively weak. Table 8 shows that both strategies substantially improve over random sampling: with only 100 validation examples, Embedding-Based and EvalTree-Based sampling outperform Random on most benchmarks, and the EvalTree-Based variant already approaches the performance obtained with the full 1273-example validation set. These results indicate that, as long as diversity and capability coverage are properly controlled, our influence-based method is not overly sensitive to validation set size, and a small but carefully constructed validation set can still provide high-quality supervision for influence estimation.

### E.5 TRAINING EFFICIENCY COMPARISON.

To further assess training efficiency, we compare our method with baseline datasets under a fixed compute budget. The dataset statistics are summarized in Table 9. Since all models are trained using the same SFT framework and identical hyperparameters, the required training FLOPs can be approximated by the total number of tokens in each dataset. As reported in Figure 11, under comparable compute budgets, the OptimSyn-based model consistently achieves the best performance among all methods, demonstrating superior training efficiency.

| Strategy | MMLU-pro | Super GPQA | Health Bench | PubMed | MedQA | HLE |
|---|---|---|---|---|---|---|
| Random | 0.5236 | 0.3016 | 0.6759 | 0.6946 | 0.5141 | 0.0942 |
| Embedding-Based | 0.5428 | 0.3138 | 0.6954 | 0.7284 | 0.5576 | 0.1036 |
| EvalTree-Based | 0.5601 | 0.3596 | 0.7434 | 0.7392 | 0.5327 | 0.0968 |
| All | 0.5697 | 0.3828 | 0.7482 | 0.8070 | 0.5875 | 0.1081 |

Table 8: Comparison of different validation sampling strategies (100-sample subsets vs. using all validation data).

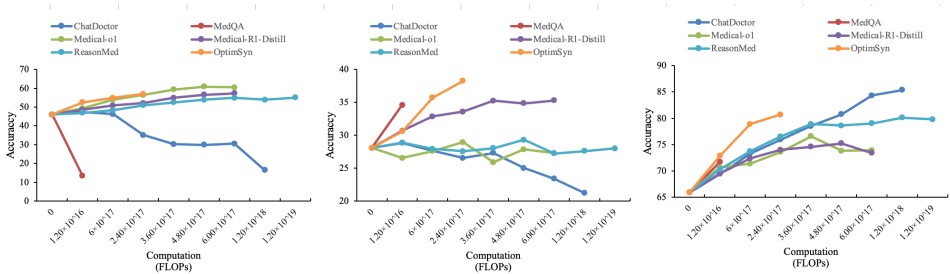

Figure 11: Comparison of SFT results under the same computational budget.

## E.6 INSTRUCTIONS FOR PROMPTER

---

**System Prompt for Prompter to Generate a Rubrics**

You are an expert in the **{domain}** field. Your core task is to instruct the "Generator" LLM to produce a single, high-quality Question–Answering (QA) data point based on the provided **{domain}** content.

---

Figure 12: Instruction template for generating a QA pair.

---

**Prompt Template for Prompter to analyze quality dimensions**

There is a given document: {document}

**Objective:**
Guide the model to generate corresponding questions and answers based on the document provided above. This QA data will be used to fine-tune a downstream Target LLM, aiming to significantly enhance its **comprehension, analysis, and creative capabilities** in {domain} tasks.

**Instructions:**

1. Randomly select **one question type** from the following set: **[Multiple choice, Fill-in-the-blank, Short answer, Essay]**.

2. Based on the characteristics of the document, analyze which quality dimensions should be prioritized for the question and answer under the selected type (with the ultimate goal of enabling the model to perform well in {domain} downstream tasks).
   The analysis may include, but is not limited to: the document's topic, style, purpose, complexity level, target audience, linguistic style, and knowledge density. Broad quality dimensions (e.g., scope, clarity, creativity, relevance, reasoning depth, factual accuracy) can be used, but their importance and descriptions should be adapted to match the specific characteristics of the document.
   At least three additional dimensions beyond the above examples should be considered — such as completeness, format compliance, tone alignment, safety, neutrality, information density, instructional precision, or adaptability to ambiguous inputs.
   When defining and prioritizing quality dimensions, you may draw on examples such as:
   **Prompt-related**: clarity, unambiguity, executability, conciseness, focus, language and grammar correctness, format appropriateness, domain relevance, bias avoidance, temporal validity, contextual coherence.
   **Response-related**: correctness, completeness, relevance, logical soundness, conciseness, fluency, credibility of information sources, creativity, information value density, neutrality, safety.
   **Prompt–Response alignment**: consistency, logical linkage, coverage of all sub-questions, appropriate disambiguation, handling of potentially incorrect prompts, compliance with constraints.
   **Technical/trainability aspects**: parsability, consistent encoding, absence of PII, diversity across samples, low noise, length suitability, copyright compliance.
   Avoid using fixed templates or rigid numerical scales. Do not provide specific content of the Q–A; instead, focus on evaluating which dimensions matter most and why.

3. The standards must be actionable to ensure the "Generator" can accurately understand and synthesize the required high-quality data.

**Output Format:**
```
{"Prompt-related": ...,
 "Response-related": ...,
 "Prompt-Response alignment": ...,
 "Technical/trainability aspects": ...}
```

Figure 13: Prompt template for analyzing quality dimensions before QA generation.

| Dataset | Tokens (M) | Num. |
|---|---|---|
| ChatDoctor | 246.3 | 100K |
| MedQA | 15.7 | 12.7k |
| Medical-o1 | 557.9 | 19.7k |
| Medical-R1-Distill | 579.4 | 22k |
| ReasonMed | 645.4 | 370k |
| OptimSyn | 125.7 | 26.6k |

Table 9: Data Statistics.

## E.7 INSTRUCTIONS FOR GENERATOR

---

**System Prompt for Generator to Generate a QA pair**

You are an expert in the field of {domain}. I will now provide you with a document. Based on this document, you need to synthesize **ONLY 1 pair** of high-quality question–answer (QA) data, which will be used for downstream task training.

---

Figure 14: Instruction template for generating a QA pair.

---

**Prompt Template for Generator to generate QA**

There is a given document: {doc}

## Specific requirements for generating the QA data:

1. Generate ONLY 1 QA pair.

2. Question Quality Standard: {Prompt-related}.

3. Answer Quality Standard: {Response-related}.

4. Prompt-Response alignment: {Prompt-Response alignment}.

5. Technical/trainability aspects: {Technical/trainability aspects}.

6. For multiple-choice questions, it is necessary to generate a "think step by step" reasoning process in the answer, ultimately leading to the conclusion.

7. All generated questions and answers must be closely related to the given content. However, do not indicate that you can see the content or use expressions like "according to the document." Do not mention the document at all.

8. The output format must be JSONL, with the specific structure as follows:

```
{
  "messages": [
    {"role":"system", "content":"You are a helpful assistent."},
    {"role":"user", "content":Question},
    {"role":"assistant", "content":Answer}
  ]
}
```

---

Figure 15: Prompt template for generating QA data in the HSS and Medical domain.

## E.8 EXAMPLES OF RUBRICS

There is a given document: **{doc}**

1. Generate ONLY 1 QA pair.
2. Question Quality Standard: {
    "**Clarity**": "Questions must be clear and unambiguous, explicitly referring to specific categories of medieval hospitals, such as small hostels/hospices, large civic hospitals, or hospitals established by crusading orders, to ensure precise alignment with the structured descriptions in the document.",
    "**Domain Relevance**": "Questions should focus on core themes such as religious character, the process of medicalization, social service recipients, and drivers of expansion, to maintain high relevance to the Humanities and Social Sciences domain.",
    "**Format Appropriateness**": "For short-answer questions, prompts should specify the required response format, such as briefly explaining the main function or expansion reason of a particular type of hospital, to facilitate evaluation."
    },
3. Answer Quality Standard: {
    "**Correctness**": "Answers must accurately reflect the definitions in the document, for example correctly describing how a type of game uses power, to ensure factual reliability.",
    "**Completeness**": "Responses should address key aspects of the game type, including its objective, primary participants, and operating mechanisms.",
    "**Relevance**": "Answers must avoid introducing unrelated theories or cases that do not pertain to organizational political games, so as to maintain focus."
    },
4. Prompt-Response alignment: {
    "**Consistency**": "The answer must remain consistent with the question's focus, such as clearly differentiating the roles and objectives of two game types when making comparisons.",
    "**Format Matching**": "For multiple-choice questions, the answer should correspond to the provided options and avoid adding new, unlisted information.",
    "**Handling Ambiguous Inputs**": "If the question is vague, the answer should explicitly reference the name of a specific game type from the document to ensure clarity."
    },
5. Technical/trainability aspects: {
    "**Parsability**": "QA pairs must use structured, machine-readable formats (such as JSON) to facilitate automated processing and training.",
    "**Copyright Compliance**": "Answers should not reproduce the text verbatim but instead paraphrase in original language while preserving meaning.",
    "**Length Suitability**": "Responses should be concise, ideally within 200 tokens, to prevent truncation during model training."
    }
6. For multiple-choice questions, it is necessary to generate a "think step by step" reasoning process in the answer, ultimately leading to the correct conclusion.
7. All generated questions and answers must be closely related to the given content. However, do not indicate that you can see the content or use expressions like "based on the document." Do not mention the document at all.
8. The output format must be JSONL, with the following structure:
    {
        "*messages*": [*{"role": "system", "content": "you are a helpful assistent."}*,
            *{"role": "user", "content": Question}*,
            *{"role": "assistant", "content": Answer}*]
    }

Figure 16: An example of generated rubrics for data synthesis.

There is a given document: **{doc}**

1. Generate ONLY 1 QA pair.
2. Question Quality Standard: Questions should have high clarity and strong domain relevance, focusing directly on the core concepts described in the source document about medieval hospitals — including their religious foundations, structural variations, medicalization process, and historical expansion. For example, they might compare differences between small hostels/hospices and large civic hospitals, analyze the role of the Crusades in hospital network expansion, or examine how institutions such as Sta Maria Nuova evolved to serve both indigent and higher-class patients. Avoid vague wording that could obscure distinctions between civic and religious institutions or the nature of services provided. Each question should clearly set the theme and scope and, when applicable, identify relevant historical categories. For short-answer formats, prompts may indicate the expected response structure (e.g., "Summarize how Sta Maria Nuova transitioned from a hospice to a medicalized hospital in two sentences"). Ideally, questions should require synthesis across the entire text rather than retrieval of a single sentence.
3. Answer Quality Standard: Answers should accurately reflect the text's historical details, distinguishing between different hospital types (religious vs civic, small hospices vs large urban hospitals), and highlighting contextual factors such as the influence of Crusading orders and non-military brotherhoods. Responses must be logically coherent, factually correct, and address all relevant aspects mentioned in the question, including chronology, institutional roles, and socio-economic reach. Information density can be enhanced by providing key examples from the text (e.g., St Leonard's in York, St Bartholomew's, Paris Hôtel Dieu, Sta Maria Nuova's medicalization). Answers must rely on a comprehensive reading of the entire passage, not just isolated lines, and preserve the distinctions made in the text. Clear differentiation between similar cases will help maintain alignment with the text's historical framework.
4. Prompt-Response alignment: Strong alignment is shown when the answer directly ties to the question's focus and cites concrete examples from the text that illustrate structural, functional, and historical contrasts. If the prompt requests a comparison of hospital types or factors driving expansion, the answer should deliver that comparison with historical examples (e.g., Crusader-built hospitals vs. Order of the Holy Spirit network). Completeness involves covering all subcomponents in the question (such as stakeholders, mechanisms, intended outcomes) and presenting them in the expected structure and tone. Matching quality should consider whether the depth, detail selection, and contextual fit align with the question's intent. Logical consistency must be maintained, avoiding conflation between distinct institutions or time periods.
5. Technical/trainability aspects: Paraphrasing should replace verbatim copying to maintain meaning while improving linguistic flexibility. Responses should avoid excessive length to stay within model context limits. A neutral, academic tone should be maintained, factually describing hospital foundations, structures, and development without subjective evaluation. These standards guide the generation of high-quality question–answer pairs that are clearly aligned with the theme of medieval hospitals, historically accurate, and evaluable in training contexts.
6. For multiple-choice questions, it is necessary to generate a "think step by step" reasoning process in the answer, ultimately leading to the correct conclusion.
7. All generated questions and answers must be closely related to the given content. However, do not indicate that you can see the content or use expressions like "based on the document." Do not mention the document at all.
8. The output format must be JSONL, with the following structure:

```
{
    "messages": [{"role": "system", "content": "you are a helpful assistent."},
        {"role": "user", "content": Question},
        {"role": "assistant", "content": Answer}]
}
```

Figure 17: An example of generated rubrics for data synthesis.

## E.9    EXAMPLES OF SYNTHETIC QA PAIRS

*System prompt:* You are a helpful assistant.

*Question*: Briefly compare the differences between large medieval civic hospitals and small hostels/hospices in terms of religious background, medical staff allocation, and drivers of expansion.

*Answer*: Large civic hospitals were typically established in urban centers, with a strong religious background, but supported by city economies and municipal funding. They gradually incorporated professional doctors, pharmacists, and auxiliary staff, achieving a high degree of medicalization, and catered to the poor while also providing private rooms for upper-class patients. In contrast, small hostels or hospices were mostly tied to local religious charity traditions, limited in scale, and had almost no resident medical staff. Their expansion was driven mainly by charitable donations from local religious orders or parishes, rather than systematic public funding.

Figure 18: An example of synthetic QA pairs.

*System prompt:* You are a helpful assistant.

*Question*: Analyze how the religious character and medicalization process of medieval hospitals were manifested in different types of institutions (such as small hostels/hospices, large civic hospitals, and hospitals established by crusading orders), and compare their differences in medical staff allocation, social service recipients, and drivers of expansion.

*Answer*: Medieval hospitals generally possessed a strong religious character. Small hostels or hospices were essentially attached to charitable religious traditions, primarily serving pilgrims with lodging and basic care in their early stages, and lacked resident medical personnel. In contrast, large civic hospitals such as Paris's Hôtel Dieu and Florence's Santa Maria Nuova gradually introduced dedicated doctors and pharmacists, signaling significant progress in medicalization. These institutions not only treated the poor but also provided private rooms reserved for upper-class patients, reflecting diversity in their social service recipients. Hospitals established by crusading orders—such as the Knights Hospitaller, the Knights Templar, and the Teutonic Knights—were mostly located in Mediterranean and German-speaking regions, primarily serving the needs of military forces and local populations along their routes, with expansion driven by military campaigns and religious missions rather than urban public service demands. The growth of small religious hostels was constrained by funding and local religious resources, while large civic hospitals relied on urban populations and municipal support, and the network of crusader hospitals was sustained by the transregional organizational strength of the orders. Overall, these three types of institutions were similar in their pervasive Christian ethos but differed sharply in the scale of medical staff deployment, the breadth of their service recipients, and the factors driving their expansion.

Figure 19: An example of synthetic QA pairs.

## E.10 EXAMPLES OF SEED DATA

---

**Title / Chapter: HOSPITALS**

Medieval hospitals were religious foundations through and through. Those planted in the West had originally been small and mainly for pilgrims; their late medieval successors were often more impressive. St Leonard's in York had 225 sick and poor in 1287; still larger were the civic hospitals of Milan, Siena and Paris. In Florence alone, a city of some 30,000 inhabitants, there were over thirty foundations by the fifteenth century. Some had only ten beds, others hundreds. In England hospitals and almshouses totalled almost five hundred by 1400 , though few were of any size or significance. London's St Bartholomew's dates from 1123 and St Thomas's from around 1215. At Bury St Edmunds six hospitals were endowed between I150 and I260 to cater for lepers, pilgrims, the infirm and the aged.\n\nSmall hospitals were essentially hostels or hospices lacking resident medical assistance, but physicians were in attendance by 1231 at the Paris Hôtel Dieu, next to Notre Dame, and Sta Maria Nuova in Florence was gradually medicalized: from twelve beds in 1288 for 'the sick and the poor', this 'first hospital among Christians', as one Florentine patriot called it, expanded by 1500 to a medical staff of ten doctors, a pharmacist and several assistants, including female surgeons. Although catering largely for the indigent, it had eight private rooms 'reserved for the sick of the higher classes'. Within hospital walls the Christian ethos was all-pervasive.\n\nIn hospital expansion the Crusades played their part, since crusading orders such as the Knights of St John of Jerusalem (later the Knights of Malta), the Knights Templar, and the Teutonic Knights built hospitals throughout the Mediterranean and German-speaking lands. By the fourteenth century non-military brotherhoods, such as the Order of the Holy Spirit, were also running infirmaries from Alsace to Poland, while the Order of St John of God appeared in Spain in the sixteenth century, building insane asylums and putting up about 200 hospitals in the New World.

---

Figure 20: An example of seed data.

