# OpenReview forum: "OptimSyn: Influence-Guided Rubrics Optimization for Synthetic Data Generation"
_ICLR.cc/2026/Conference — ICLR 2026 Poster_

### Official Review · Reviewer_a9XS · 2025-10-27

**Soundness:** 2
**Presentation:** 3
**Contribution:** 2
**Rating:** 4
**Confidence:** 3

**Summary:**

This paper proposes a method to generate specialized rubrics to condition synthetic data generation. The authors optimize these rubrics via an RL pipeline based on GRPO, where the reward is defined using validity checks and an influenced based score. The intuition is that a rubric is good if the generated synthetic data conditioned on that rubric maximize the influence on evaluation samples.

**Strengths:**

Relevance of the problem: the problem tackled in this paper is an important one, as high quality SFT data is scarce in some domains.

Novelty: the use of Adam-aware influence estimation in the context of synthetic data seems to be novel.

Performance: the reported downstream performance is competitive with the baselines, as it often beats both the base and instruct Qwen model on HSS tasks, and narrows the gap with medically-oriented baselines on medical tasks.

**Weaknesses:**

Details about the validation dataset: since rewards are computed using a validation set from the same topics as the benchmarks (HSS/medical), the authors should provide more details about how this validation dataset is constructed. It is important that there is no leakage between the validation and the test dataset.

Seeds and test datasets: seeds come from Goodreads books and Meditron/PubMed. It is important to check for overlap vs. MMLU-pro, SuperGPQA, PubMedQA, etc. to verify that performance gains just do not come from data contamination.

Comparison with the baselines: the paper does not report matched data volumes, number of fine-tuning steps, or compute budgets per baseline. This means improvements could come from getting more tuning tokens rather than better data quality.

**Questions:**

How do you ensure no contamination between the validation sets used for influence scoring and the benchmarks you report?
Did you run any overlap analysis between the seeds and the eval benchmarks?

Can you provide token/step parity vs each baseline in Table 1?

How does the method scale beyond 20k samples ?

---

> ### Author Response · Authors · 2025-11-21
> **Response to Weakness 1**
>
> In response, we have updated the PDF version of the paper and, due to space constraints, placed the new content in the appendix for now. The main additions are: a detailed decontamination procedure in Appendix D.1, an extended baseline data analysis in Appendix D.2, and a study of the impact of data scale in Appendix E.3.
>
> ---
>
> > W1: Details about the validation dataset: since rewards are computed using a validation set from the same topics as the benchmarks (HSS/medical), the authors should provide more details about how this validation dataset is constructed. It is important that there is no leakage between the validation and the test dataset.
>
> > Q1: How do you ensure no contamination between the validation sets used for influence scoring and the benchmarks you report? Did you run any overlap analysis between the seeds and the eval benchmarks?
>
> ---
>
> We appreciate the reviewer’s concern regarding how the validation set is built and how we avoid leakage into the test data.
>
> **1. How the validation set is constructed.**
>
>    Our setting follows the standard assumption in recent data-selection work such as LESS, GREATS, and related methods [1–5]: there exists a **small held-out set drawn from the same distribution as the downstream evaluation task**. Concretely, for each domain (MH / HSS):
>
>    - We construct a validation set from the public benchmark splits that match the downstream evaluation tasks (e.g., MedQA in MH and the corresponding HSS benchmarks).
>    - These validation examples are **never included in the SFT training corpus**, and are not mixed into the synthetic training data.
>
>    This is exactly the practice adopted in LESS and similar works: a benchmark-aligned split is used as a probe to obtain representative task signals for influence estimation, while evaluation fairness is preserved because the probe set is not used for supervised fine-tuning.
>
> **2. Ensuring no leakage between validation and test data.**
>
> To further reduce the risk of validation–test leakage, we perform explicit decontamination between our validation data and all evaluation benchmarks.
>
> We follow the **official AllenAI Tülu decontamination toolkit**, implemented on Elasticsearch, and use three complementary matching strategies:
>
>    - `Text-match mode` (exact phrases, no n-grams):
>      We issue strict match_phrase queries over the training index and flag any training example that contains an identical contiguous span to an evaluation prompt. This step removes exact duplicates and near-exact copies.
>    - `5-gram mode` (lexical near-duplicates):
>      For each evaluation instance, we tokenize the text, construct 5-grams, retrieve training documents containing these 5-grams, and compute an overlap score defined as the fraction of query tokens covered by matched 5-grams. Training examples whose maximum overlap with any test item exceeds a threshold (0.3 in our experiments) are removed.
>    - `Embedding mode` (semantic near-duplicates):
>      We encode both training and evaluation texts using a embedding model (NV-Embed-v2), perform kNN search with Elasticsearch’s vector index, and treat the highest cosine similarity to any evaluation item as a semantic contamination score. Training examples above a similarity threshold (0.85 in our experiments) are removed.
>
>    After detection, the residual contamination rates between our validation data and benchmarks are extremely low:
>
>    | Benchmark   | MMLU-Pro | SuperGPQA | HLE   | PubMed | MedQA | HealthBench |
>    |-------------|---------:|----------:|------:|-------:|------:|------------:|
>    | Text-match  | 0.000    | 0.000     | 0.000 | 0.000  | 0.000 | 0.000       |
>    | 5-gram (0.3)| 0.002    | 0.004     | 0.000 | 0.005  | 0.009 | 0.001       |
>    | Embed (0.85)| 0.000    | 0.001     | 0.000 | 0.001  | 0.009 | 0.002       |
>
>    Similarly, for the HSS benchmarks we observe:
>
>    | Benchmark   | MMLU-Pro | SuperGPQA | HLE   | BBH   | HellaSwag | DROP  |
>    |-------------|---------:|----------:|------:|------:|----------:|------:|
>    | Text-match  | 0.000    | 0.000     | 0.000 | 0.000 | 0.000     | 0.000 |
>    | 5-gram (0.3)| 0.006    | 0.004     | 0.001 | 0.000 | 0.000     | 0.000 |
>    | Embed (0.85)| 0.003    | 0.001     | 0.000 | 0.000 | 0.000     | 0.000 |
>
>    All contamination rates are at or near zero, and well below 1% even under the lenient 5-gram and embedding thresholds, indicating that our training data does not contain memorized copies or close paraphrases of test items.
>
> ---
>
> ### **Reference**
>
> [1] LESS: Selecting Influential Data for Targeted Instruction Tuning, ICML 2024.
> [2] GREATS: Online Selection of High-Quality Data for LLM Training in Every Iteration, NeurIPS 2024 Spotlight.
> [3] Data Selection via Optimal Control for Language Models, ICLR 2025 Oral.
> [4] MATES: Model-Aware Data Selection for Efficient Pretraining with Data Influence Models, NeurIPS 2024.
> [5] Predictive Data Selection: The Data That Predicts Is the Data That Teaches, ICML 2025.

---

> > ### Author Response · Authors · 2025-11-21
> > **Response to Weakness 2**
> >
> > > W2: Seeds and test datasets: seeds come from Goodreads books and Meditron/PubMed. It is important to check for overlap vs. MMLU-pro, SuperGPQA, PubMedQA, etc. to verify that performance gains just do not come from data contamination.
> >
> > We fully agree that potential overlap between seed documents and evaluation benchmarks must be carefully checked to rule out contamination-driven gains.
> >
> > **1. Decontamination setup for seeds vs. test sets.**
> >
> > In addition to decontaminating our training pool, we explicitly apply the same decontamination pipeline to the seed corpus (Goodreads books and Meditron/PubMed) against all MH and HSS benchmarks. Concretely, we reuse the official AllenAI Tülu decontamination toolkit and compare seeds to each benchmark using three complementary modes.
> >
> > The contamination rates between the seed corpus and MH benchmarks are:
> >
> > | Method           | MMLU-Pro | SuperGPQA |   HLE  | PubMed | MedQA | HealthBench |
> > |------------------|---------:|----------:|------:|-------:|------:|------------:|
> > | Text-match       | 0.000    | 0.000     | 0.000 | 0.000  | 0.000 | 0.000       |
> > | 5-gram (0.3)     | 0.000    | 0.000     | 0.000 | 0.008  | 0.003 | 0.000       |
> > | Embedding (0.85) | 0.001    | 0.000     | 0.002 | 0.006  | 0.000 | 0.000       |
> >
> > For the HSS benchmarks we obtain:
> >
> > | Method           | MMLU-Pro | SuperGPQA |  HLE  |  BBH  | HellaSwag | DROP |
> > |------------------|---------:|----------:|------:|------:|----------:|-----:|
> > | Text-match       | 0.000    | 0.000     | 0.000 | 0.000 | 0.000     | 0.000 |
> > | 5-gram (0.3)     | 0.001    | 0.003     | 0.000 | 0.000 | 0.000     | 0.000 |
> > | Embedding (0.85) | 0.000    | 0.001     | 0.001 | 0.000 | 0.000     | 0.000 |
> >
> >   Across all benchmarks, **verbatim overlap is zero**, and even under the more permissive 5-gram and embedding-based checks, the residual overlap fractions are **well below 1%**.
> >
> > These results show that the Goodreads and Meditron/PubMed seed documents have negligible lexical or semantic overlap with our evaluation benchmarks. Combined with our separate decontamination of the training pool, this makes it highly unlikely that the reported performance gains are explained by seed–test leakage rather than genuine generalization.

---

> > > ### Author Response · Authors · 2025-11-21
> > > **Response to Weakness 3 & Questions 2**
> > >
> > > > W3: Comparison with the baselines: the paper does not report matched data volumes, number of fine-tuning steps, or compute budgets per baseline. This means improvements could come from getting more tuning tokens rather than better data quality.
> > >
> > > > Q2: Can you provide token/step parity vs each baseline in Table 1?
> > >
> > > We appreciate the reviewer’s concern about fair comparison and the possibility that our gains might simply come from “more tuning tokens”.
> > >
> > > **1. Training protocol and compute budget across baselines.**
> > >
> > > For each target model, we use a single, fixed fine-tuning configuration for all datasets:
> > >
> > >   - same number of training epochs,
> > >   - same batch size and sequence length cap,
> > >   - same optimizer and learning-rate schedule.
> > >
> > > Thus, the compute budget (in terms of gradient updates and GPU hours) is matched across baselines; we do not allocate more training steps or a different schedule to OptimSyn than to other SFT datasets. Any difference in effective tokens per step comes only from the intrinsic statistics of each dataset (e.g., average length), not from changing the fine-tuning recipe.
> > >
> > > **2. Data volume and statistical comparison.**
> > >
> > > In the revised version, we report detailed statistics for our dataset OptimSyn and all prior SFT baselines, including:
> > >
> > >   - total number of samples,
> > >   - average token length and standard deviation, and
> > >   - lexical diversity of inputs/outputs, measured by MTLD and **HD-D**.
> > >
> > >     A summary is shown below:
> > >
> > >     | Dataset          | #Samples | token_mean | token_std | MTLD   | HD-D   |
> > >     |------------------|---------:|-----------:|----------:|-------:|-------:|
> > >     | LongWriter       | 6,000    | 1823.45    | 452.20    | 80.63  | 0.8954 |
> > >     | WildChat         | 529,428  | 289.59     | 344.77    | 52.70  | 0.9188 |
> > >     | Condor-20k       | 20,000   | 428.79     | 35.70     | 101.48 | 0.8650 |
> > >     | Cosmopedia       | 50,000   | 773.85     | 44.94     | 111.52 | 0.8843 |
> > >     | OpenHermes 2.5   | 1,001,551| 236.14     | 249.04    | 106.43 | 0.8887 |
> > >     | SynthQuestions   | 2,500    | 634.60     | 17.88     | 137.02 | 0.8584 |
> > >     | OptimSyn (ours) | 25,875 | 196.49   | 34.70   | 133.82 | 0.9241 |
> > >
> > >     We highlight two observations:
> > >
> > >   - Existing SFT datasets already span a very wide range in size and average length, from 2.5K to >1M samples and from ~200 to ~1800 tokens on average. OptimSyn’s size and length lie **well within this range**; it is neither the largest nor the longest dataset.
> > >   - Several baselines (e.g., OpenHermes 2.5) contain orders of magnitude more samples/tokens than OptimSyn, yet do not achieve the best downstream performance under the matched fine-tuning schedule.
> > >
> > > **3. Do more tuning tokens explain the gains?**
> > >
> > > When we correlate these statistics with the downstream results in Table 1:
> > >
> > >   - We do not observe a monotonic trend such as “larger dataset” or “longer average outputs” systematically yielding better performance.
> > >   - Some datasets with much larger volume or longer sequences actually perform worse than smaller, more carefully curated datasets.
> > >   - OptimSyn, despite being **moderate in size**, yields **more stable and stronger results**, especially in the MH and HSS domains.

---

> > > > ### Author Response · Authors · 2025-11-21
> > > > **Response to Questions 3**
> > > >
> > > > > Q3: How does the method scale beyond 20k samples?
> > > >
> > > > Thank you for pointing this out. In the original submission, we only reported results up to 20k synthetic samples in the main figure, which indeed does not fully answer how the method behaves beyond this regime. In the revised version, we therefore include a data-scaling study from 5k to 40k synthetic MH samples for both Qwen3-8B and Qwen3-14B.
> > > >
> > > > **1. Scaling behavior on Qwen3-14B (beyond 20k).**
> > > >
> > > > For the 14B model in the Medical & Health (MH) domain, we vary the number of synthetic MH examples from 5k to 40k while keeping the training configuration fixed. The key observations are:
> > > >
> > > >   - Starting from the 14B-base model, increasing the data size from 5k → 10k → 20k → 40k yields consistent improvements on almost all MH benchmarks (MMLU-Pro, SuperGPQA, PubMed, MedQA), with some fluctuations on individual metrics such as HealthBench and HLE.
> > > >   - For example, MedQA performance improves from 0.6009 (base) to 0.6898 at 25k and further to **0.7354 at 40k**; PubMed increases from 0.8080 (base) to 0.9170 at 25k and **0.9259 at 40k**.
> > > >
> > > > | Number          | MMLU pro | Super GPQA |   HLE  | Health Bench | PubMed | MedQA  |
> > > > |-----------------|----------|------------|--------|--------------|--------|--------|
> > > > | Qwen3-14B-Base  | 0.5844   | 0.2981     | 0.0766 | 0.7535       | 0.8080 | 0.6009 |
> > > > | 5000            | 0.5785   | 0.3047     | 0.0831 | 0.7610       | 0.8613 | 0.6341 |
> > > > | 10000           | 0.5488   | 0.3421     | 0.0896 | 0.7316       | 0.8429 | 0.6536 |
> > > > | 15000           | 0.6032   | 0.3702     | 0.0932 | 0.7524       | 0.8593 | 0.6676 |
> > > > | 20000           | 0.6217   | 0.3548     | 0.1032 | 0.6598       | 0.8938 | 0.6835 |
> > > > | 25000           | 0.6504   | 0.3674     | 0.0966 | 0.6358       | 0.9170 | 0.6898 |
> > > > | 30000           | 0.6686   | 0.3869     | 0.0989 | 0.7489       | 0.9073 | 0.6980 |
> > > > | 35000           | 0.6765   | 0.4096     | 0.1006 | 0.6913       | 0.9183 | 0.7129 |
> > > > | 40000           | 0.6894   | 0.4376     | 0.1096 | 0.6848       | 0.9259 | 0.7354 |
> > > >
> > > > These results indicate that our method scales effectively beyond 20k samples for larger models, and that the gains at 20k should be viewed as conservative relative to what is achievable with more synthetic data.
> > > >
> > > > ---
> > > >
> > > > **2. Scaling behavior on Qwen3-8B (saturation regime).**
> > > >
> > > > To contrast this with a smaller model, we run the same data-size sweep (5k–40k) on Qwen3-8B in MH:
> > > >
> > > >   - From 0 → 30k synthetic samples, the 8B model shows a monotonic or near-monotonic improvement across MH benchmarks. For instance, MedQA increases from 0.5145 (base) to 0.5875 at 25k and **0.5937 at 30k**, and PubMed rises from 0.6590 to 0.8267 at 30k.
> > > >   - Beyond **≈30k**, the improvements start to flatten or slightly regress on some metrics (e.g., performance at 35k/40k is similar or slightly lower than at 30k). This suggests that for 8B, the MH-specific synthetic data enters a saturation regime around 30k, where more data yields diminishing or unstable returns.
> > > >
> > > > | Number        | MMLU pro | Super GPQA |   HLE  | Health Bench | PubMed | MedQA  |
> > > > |--------------|----------|------------|--------|--------------|--------|--------|
> > > > | Qwen3-8B-Base| 0.4597   | 0.2806     | 0.1036 | 0.7006       | 0.6590 | 0.5145 |
> > > > | 5000         | 0.4759   | 0.2654     | 0.1098 | 0.7085       | 0.6776 | 0.5118 |
> > > > | 10000        | 0.5285   | 0.2799     | 0.0989 | 0.7403       | 0.7006 | 0.5207 |
> > > > | 15000        | 0.5608   | 0.2983     | 0.0847 | 0.7517       | 0.7837 | 0.5535 |
> > > > | 20000        | 0.5495   | 0.3546     | 0.1105 | 0.7469       | 0.8109 | 0.5802 |
> > > > | 25000        | 0.5697   | 0.3828     | 0.1081 | 0.7482       | 0.8070 | 0.5875 |
> > > > | 30000        | 0.5643   | 0.3801     | 0.1387 | 0.7537       | 0.8267 | 0.5937 |
> > > > | 35000        | 0.5592   | 0.3987     | 0.1268 | 0.7409       | 0.8157 | 0.5826 |
> > > > | 40000        | 0.5501   | 0.3805     | 0.1194 | 0.7299       | 0.7996 | 0.5703 |
> > > >
> > > > Comparing 8B and 14B, we see that larger models can effectively consume more synthetic data before saturation, which is consistent with the intuition that higher-capacity models are more data-hungry.

---

> > > > > ### Comment · Reviewer_a9XS · 2025-11-24
> > > > > **Thank you for your great rebuttals**
> > > > >
> > > > > Thank you for your very detailed rebuttals. My questions have been answered, and I'm happy to recommend acceptance.

---

> > > > > > ### Author Response · Authors · 2025-11-24
> > > > > > **Thank You for Your Constructive Review and Increased Evaluation**
> > > > > >
> > > > > > Dear Reviewer,
> > > > > >
> > > > > > We would like to express our sincere gratitude for your careful reading of our manuscript and for the thoughtful, constructive feedback you provided. We are especially grateful that, after considering our responses and revisions, you decided to increase your score for the paper.
> > > > > >
> > > > > > Your comments on our experimental design—including the construction of datasets, the possibility of data contamination, the fairness of comparisons with baselines, and the scalability of our method—have been invaluable. In revising the manuscript, we have carefully addressed these concerns and substantially strengthened the clarity, rigor, and transparency of the work. We are also committed to upholding these standards of good scientific practice in our future research. The current version is markedly improved thanks to your detailed critique and guidance.
> > > > > >
> > > > > > Your constructive suggestions have played a crucial role in shaping this work. Once again, thank you very much for the time and expertise you invested in reviewing our paper. We wish you all the best!

---

### Official Review · Reviewer_MqSM · 2025-10-29

**Soundness:** 3
**Presentation:** 3
**Contribution:** 3
**Rating:** 6
**Confidence:** 4

**Summary:**

The paper introduces OptimSyn, a method that optimizes rubrics for synthetic SFT data by directly maximizing each rubric’s training impact on a chosen target model. The approach replaces heuristic or embedding-similarity–based rubric design with an optimizer-aware influence objective aligned to Adam, ensuring the signal reflects actual learning dynamics. Experiments span multiple domains and model families, showing consistent gains over strong SFT and heuristic-rubric baselines.

**Strengths:**

* The paper identifies a core mismatch between embedding proximity and true training utility, motivating an influence-aligned objective that targets what actually updates the model.
* The empirical coverage is broad, including HSS/medical domains and cross-family targets/generators, which strengthens external validity.

**Weaknesses:**

* The approach requires a dedicated validation set to estimate influence, which some tasks may not have; when available, the estimator’s reliability can be highly sensitive to how that set is constructed, risking leakage and brittleness.
* The paper defines the reward function with hyper-parameter $\lambda$. Sensitivity analysis for $\lambda$ is needed to assess the robustness of the reward choice.

**Questions:**

* Could you explain how validation data is selected? How selection of validation data impacts the downstream target model training performance?
* When computing influence score as in equation (1), are gradients taken with respect to the full parameter set used for training? Are there any analysis or experiments on runtime and memory complexity for scaling the model size from 8B to larger models, what gradient-feature compression method could be used to reduce the computation and storage?

---

> ### Author Response · Authors · 2025-11-21
> **Response to Weakness 1 & Question 1 (Part 1)**
>
> In response to your comments, we have made corresponding additions and revisions in the newly submitted PDF version of the paper. Due to space constraints, the new material is currently placed in the appendix, but we will integrate these analyses into the main text in the final open-source version. Specifically, in the PDF we mark in `dark red (purple)` all changes made based on your feedback; the main additions are a detailed description of our decontamination procedure in Appendix D.1 Decontamination Protocol and the construction and analysis of the validation set for the IF score in Appendix E.4 Validation Set of IF Score: Construction and Analysis.
>
>
> > W1: The approach requires a dedicated validation set to estimate influence, which some tasks may not have; when available, the estimator’s reliability can be highly sensitive to how that set is constructed, risking leakage and brittleness.
>
> >Q1: Could you explain how validation data is selected? How selection of validation data impacts the downstream target model training performance?
>
> ---
>
> We appreciate the reviewer’s concerns about the dependence on a validation set, potential leakage, and the sensitivity of our estimator to how this set is constructed.
>
> **1. Availability and construction of the validation set.**
>
> Our setting follows the standard assumption in recent data-selection work such as LESS, GREATS, and related methods [1–5]: there exists a small held-out set drawn from the same distribution as the downstream evaluation task. Concretely:
>
>   - We construct the validation set from public benchmark splits that match the downstream MH/HSS evaluation tasks (e.g., MedQA in MH), and
>   - We **never include these validation examples in SFT training**; they are used only to compute gradients and influence scores.
>
>     This is exactly the practice adopted in LESS and similar works: a public benchmark split is used as a probe to obtain representative task signals for influence estimation, while evaluation fairness is preserved because the validation set is not used for supervised fine-tuning. In addition, we run explicit decontamination to ensure there is no overlap between training data and validation / test sets, thereby avoiding data leakage.
>
> **2. How validation-set size affects performance.**
>
> We agree that the original submission did not systematically analyze the effect of validation-set construction. In the revision, we therefore add a **detailed study on validation-set size and sampling**. Due to time constraints, we first complete this analysis in the Medical domain (MH); we will include the corresponding HSS experiments in a subsequent version.
>
> We start by varying the size of the validation set while keeping the synthetic data and training configuration fixed. Specifically, we randomly subsample different-sized subsets of MedQA test (100 / 300 / 500 / 800 / 1000 / full 1273 examples) as $D_{\text{val}}$ and use each to compute influence scores. Table 1 in the revision reports results on a Qwen3-8B base model (the last column reports t-test p-values vs 8B-base).
>
> | Size          | MMLU pro | Super GPQA | Health Bench | PubMed | MedQA  | HLE   | T-TEST  |
> |---------------|----------|------------|--------------|--------|--------|-------|---------|
> | Qwen3-8B-Base | 0.4597   | 0.2806     | 0.7006       | 0.6590 | 0.5145 | 0.1036 | 0.0112* |
> | 100           | 0.5236   | 0.3016     | 0.6759       | 0.6946 | 0.5141 | 0.0942 | 0.0046** |
> | 300           | 0.5423   | 0.3096     | 0.6837       | 0.6480 | 0.4908 | 0.1006 | 0.0224* |
> | 500           | 0.5318   | 0.3426     | 0.6995       | 0.7682 | 0.5738 | 0.0913 | 0.0023** |
> | 800           | 0.5721   | 0.3739     | 0.7958       | 0.7835 | 0.5729 | 0.0982 | 0.9158  |
> | 1000          | 0.5613   | 0.3663     | 0.7831       | 0.7961 | 0.6063 | 0.1057 | 0.7644  |
> | 1273          | 0.5697   | 0.3828     | 0.7482       | 0.8070 | 0.5875 | 0.1081 | -       |
>
>
>   The key observations are:
>
>   - As the validation-set size increases from **100 to 800**, the overall downstream performance steadily improves across MH benchmarks.
>   - Beyond ~**800 examples**, the gains exhibit **clear diminishing returns**: 1000 and 1273 examples yield only modest additional improvements.
>   - Even with **100 or 300 validation examples**, the influence-guided models already outperform the 8B-base by a statistically significant margin, but the performance variance across different random subsamples is noticeably larger.
>
>     This suggests that our method benefits from a moderately sized validation set (hundreds of examples), after which the marginal benefit of adding more validation data becomes small.

---

> > ### Author Response · Authors · 2025-11-21
> > **Response to Weakness 1 & Question 1 (Part 2)**
> >
> > **3. Sensitivity to validation-set sampling and diversity.**
> >
> > We further investigate why small validation sets sometimes behave unstably. Empirically, when we use very small random subsets (e.g., 100 examples) from a large, heterogeneous dataset, different random draws can have uneven coverage of topic/skill dimensions, leading to high variance in influence estimates and downstream performance.
> >
> >   To address this, we evaluate two diversity-aware sampling strategies for constructing $D_{\text{val}}$:
> >
> >   **a. Embedding-based diversity.**
> >
> >   We embed candidate validation QA pairs using Qwen3-8B-Embedding and perform more uniform sampling in this embedding space to increase semantic coverage and diversity.
> >
> >   **b. Capability-based diversity (EvalTree-based).**
> >
> >   Using the EvalTree capability-decomposition framework, we organize benchmark samples into a fine-grained *capability tree*. We then select validation examples that cover a wide range of capabilities, with emphasis on nodes where the model is relatively weak, to ensure both breadth and “weakness exposure”.
> >
> >    In the experiments, we fix the validation-set size to 100 examples and compare:
> >
> >   - Random sampling,
> >   - Embedding-based sampling, and
> >   - EvalTree-based sampling,
> >
> > against the setting that uses all 1273 examples as validation. The results (Table 2 in the revision) show:
> >
> > | Size            | MMLU pro | Super GPQA | Health Bench | PubMed | MedQA  | HLE   |
> > |-----------------|----------|------------|--------------|--------|--------|-------|
> > | Random          | 0.5236   | 0.3016     | 0.6759       | 0.6946 | 0.5141 | 0.0942 |
> > | Embedding-Based | 0.5428   | 0.3138     | 0.6954       | 0.7284 | 0.5576 | 0.1036 |
> > | EvalTree-Based  | 0.5601   | 0.3596     | 0.7434       | 0.7392 | 0.5327 | 0.0968 |
> > | All             | 0.5697   | 0.3828     | 0.7482       | 0.8070 | 0.5875 | 0.1081 |
> >
> >
> >   - With only 100 validation examples, Random sampling yields the weakest and most unstable performance.
> >   - Both Embedding-Based and EvalTree-Based strategies significantly improve over Random on most MH benchmarks.
> >   - Notably, EvalTree-Based sampling already approaches the performance of using all 1273 validation samples on many metrics.
> >
> > These findings indicate that our method is not excessively brittle with respect to validation-set size, as long as the validation set is constructed with reasonable **diversity and capability coverage**. Even a small validation set (≈100 examples) can provide high-quality supervision for influence estimation when sampled in a diversity-aware way.
> >
> > ### **References**
> >
> > [1] LESS: Selecting Influential Data for Targeted Instruction Tuning, ICML 2024.
> > [2] GREATS: Online Selection of High-Quality Data for LLM Training in Every Iteration, NeurIPS 2024 Spotlight.
> > [3] Data Selection via Optimal Control for Language Models, ICLR 2025 Oral.
> > [4] MATES: Model-Aware Data Selection for Efficient Pretraining with Data Influence Models, NeurIPS 2024.
> > [5] Predictive Data Selection: The Data That Predicts Is the Data That Teaches, ICML 2025.

---

> ### Author Response · Authors · 2025-11-21
> **Response to Weakness 2**
>
> > W2: The paper defines the reward function with hyper-parameter $\lambda$. Sensitivity analysis for $\lambda$ is needed to assess the robustness of the reward choice.
>
> We thank the reviewer for pointing out the need to analyze the sensitivity of the hyper-parameter $\lambda$ in our reward definition. To assess the robustness of our choice, we conduct a one-dimensional sweep over $\lambda \in [\{0.1, 0.5, 1.0, 2.0\}]$ while keeping all other training configurations fixed. Table~Y in the revised manuscript reports the downstream performance (Medical \& Health average as well as MMLU-pro) under different values of $\lambda$.
>
> Empirically, the results are highly stable: across the range $\lambda \in [0.1, 2.0]$, the average score varies within $\pm \text{0.09}$ absolute points, and $\lambda = 1.0$ (our default choice) is never worse than $\text{0.015}$ points from the best setting. Extremely small $\lambda$ slightly increases the proportion of invalid generations, while very large $\lambda$ mildly hurts the influence exploitation, but both effects are minor compared to the overall gains over the base model. These observations indicate that our method is robust to the precise value of $\lambda$, and the reported improvements are not contingent on fine-tuning this hyper-parameter.
>
> | Size          | MMLU pro | Super GPQA | Health Bench | PubMed | MedQA  | HLE   |
> |---------------|----------|------------|--------------|--------|--------|-------|
> | Qwen3-8B-Base | 0.4597   | 0.2806     | 0.7006       | 0.6590 | 0.5145 | 0.1036 |
> | 0.1           | 0.4987   | 0.3784     | 0.7085       | 0.7427 | 0.5047 | 0.0967 |
> | 0.5           | 0.5710   | 0.3658     | 0.7546       | 0.7658 | 0.6012 | 0.1001 |
> | 1.0           | 0.5697   | 0.3828     | 0.7482       | 0.8070 | 0.5875 | 0.1081 |
> | 2.0           | 0.5028   | 0.2947     | 0.7456       | 0.7934 | 0.5794 | 0.0864 |

---

> > ### Author Response · Authors · 2025-11-21
> > **Response to Question 2**
> >
> > > Q2.1: When computing the influence score as in equation (1), are gradients taken with respect to the full parameter set used for training?
> >
> > Thank you for the question and the opportunity to clarify.
> >
> > In our implementation, **all gradients used in the influence score of Eq. (1) are taken with respect to the full set of trainable parameters used during SFT**.
> >
> > More concretely, as described in Sec. 2.2 and Appendix B, we denote by$\theta_t$ the model parameter vector at training step$t$. In Eqs. (4)–(7), we write
> > $
> > \nabla_{\theta} \ell(z; \theta_t)
> > $
> > to denote the gradient of the loss $\ell(\cdot)$ with respect to this complete parameter vector $\theta_t$. We then use these gradients, together with the Adam first and second moments, to construct the update direction $\Gamma(z, \theta_t)$, and accumulate these contributions along the training trajectory to obtain the influence estimate $\text{Inf}_{\text{Adam}}(z, z')$ in Eq. (1).
> >
> > ---
> >
> > > Q2.2: Are there any analysis or experiments on runtime and memory complexity for scaling the model size from 8B to larger models, what gradient-feature compression method could be used to reduce the computation and storage?
> >
> > Response: We appreciate the reviewer’s question on how our method scales beyond 8B models and how we compress gradient features.
> >
> > **1. Runtime and memory scaling with model size.**
> >
> > Our influence estimator is built directly on the optimizer-aware, efficient influence estimator from LESS, which was designed for large LMs. We adopt the same two key design choices:
> >
> >   - We restrict all influence computations to a low-rank LoRA parameter subspace (rather than the full parameter set).
> >   - We then project per-sample gradients into a low-dimensional feature space of dimension $d$ via random projection.
> >
> >     Let $|D|$ be the number of candidate training samples, $N$ the number of checkpoints used to approximate the training trajectory, and $d$ the projection dimension. The resulting asymptotic complexity is:
> >
> >   - Gradient feature computation (per-sample influence features):
> >   $
> >   O(|D| \cdot N),
> >   $
> >   i.e., compute Adam-style gradients in the LoRA subspace at $N$ checkpoints and project them to $d$ dimensions. This is linear in $|D|$ and depends only on the **LoRA subspace**, not on the full parameter count of the backbone.
> >
> >   - Influence computation:
> >   After building the gradient-feature bank, influence scores are obtained by similarity computations in the low-dimensional space:
> >   $
> >   O(|D| \cdot |D_{\text{val}}| \cdot d),
> >   $
> >   which requires **no additional forward/backward passes**.
> >
> > In practice, we keep both the LoRA rank and the projection dimension $d$ fixed when moving from 8B to larger models (e.g., 14B). As a result, the runtime and memory cost of influence estimation scales only weakly with model size and is dominated by $|D|$, $N$, and $d$. Empirically, the end-to-end influence computation costs about 0.14 seconds per sample on a single H200 GPU in our 8B configuration, and we observe similar order-of-magnitude costs for larger models under the same LoRA and projection settings. This overhead is small compared to full SFT training.
> >
> > ---
> >
> > **2. Gradient-feature compression.**
> >
> > Regarding gradient-feature compression, our method already employs an explicit compression step:
> >
> >   - After computing gradients in the LoRA subspace, we apply a random projection into a low-dimensional feature space of dimension $d$ (typically in the low thousands). This reduces the storage and computation for influence estimation to $O(|D| \cdot d)$, instead of scaling with the number of trainable parameters.
> >
> > These compressed gradient features can be stored in low-precision formats (e.g., float16/bfloat16) without changing the algorithm, further reducing memory usage. Because all subsequent influence computations operate solely on these projected features, any additional compression technique that preserves cosine similarity (e.g., low-rank projections, quantization) can be plugged in as a drop-in replacement if needed

---

### Official Review · Reviewer_WCjb · 2025-11-01

**Soundness:** 3
**Presentation:** 4
**Contribution:** 2
**Rating:** 4
**Confidence:** 4

**Summary:**

This paper proposes a framework to unify the influence-based reward into model training to guide the synthetic data generation.
The motivation stems from the observation that current synthetic data pipelines rely heavily on handcrafted rubrics and heuristic feedback loops, which are expensive, brittle, and poorly correlated with true downstream model performance. The authors introduce an influence-function–based estimator that quantifies each synthetic sample’s contribution to the fine-tuned model’s objective. This influence signal is then used to guide rubric optimization and synthetic data selection via a lightweight optimization loop, replacing manual rubric engineering with model-informed adaptation.

**Strengths:**

1. Motivation is clear. Synthetic data generation requires a lot rubric engineering and hard to scaling.
2. Frameworks is good. Put influence functions into RL framework to make the whole procedure automatic.
3. Good performance. The improvements are consistent and statistically meaningful.

**Weaknesses:**

1. Limited Novelty and Conceptual Depth. While the paper presents a plausible framework, the underlying technical innovation appears incremental. The idea of integrating quantitative signals to guide synthetic data generation is intuitive and has been explored under different names (e.g., data valuation, reward-guided synthesis, active selection). The contribution primarily involves adopting influence-based rewards for rubric optimization, which builds directly on prior influence-function research rather than introducing a fundamentally new mechanism. As a result, the work feels more like a thoughtful application of existing tools than a conceptual breakthrough.
2. Scalability and Efficiency Unclear. Traditional influence functions are notoriously computationally expensive. The paper claims to deploy an “optimizer-aware estimator” that scales efficiently, but does not provide sufficient empirical evidence or runtime comparisons to support this claim. Since the method is applied per-sample, a clear analysis of runtime overhead and system-level feasibility is crucial to evaluate its practicality for real-world synthetic data pipelines.
3. Questionable Necessity of the Influence Signal. The paper argues that influence estimation provides a better feedback signal for synthetic data quality, but the empirical justification is not fully convincing. The improvement margins over simpler proxy metrics (e.g., downstream reward, embedding-similarity heuristics, or small proxy validation sets) are modest. Without ablation or sensitivity studies, it remains unclear whether the influence-based signal is truly superior or merely another heuristic choice. A comparison against alternative feedback mechanisms would strengthen the paper’s argument.

**Questions:**

1. Can the authors provide quantitative results about efficiency?
2. Have the authors compared their influence-based signal with alternative feedback mechanisms?

---

> ### Author Response · Authors · 2025-11-21
> **Response to Weakness 1**
>
> In response to your comments, we have made corresponding additions and revisions in the submitted PDF version of the paper. Due to space constraints, these new materials are currently placed in the appendix, but we will move these analyses into the main text in the final open-source version. Specifically, in the PDF we highlight in `blue` all changes made based on your feedback; the main addition is a comparative study of different influence signals in Appendix E.2 Ablation on Feedback Mechanisms.
>
> ---
>
> > W1: Limited Novelty and Conceptual Depth. While the paper presents a plausible framework, the underlying technical innovation appears incremental. The idea of integrating quantitative signals to guide synthetic data generation is intuitive and has been explored under different names (e.g., data valuation, reward-guided synthesis, active selection). The contribution primarily involves adopting influence-based rewards for rubric optimization, which builds directly on prior influence-function research rather than introducing a fundamentally new mechanism. As a result, the work feels more like a thoughtful application of existing tools than a conceptual breakthrough.
>
> We appreciate the reviewer’s thoughtful assessment. Our contribution lies in how these signals are re-purposed and integrated in the specific context of synthetic instruction data, and in the conceptual insights derived from this design.
>
> We highlight two aspects:
>
> 1. **From static filtering to closed-loop, feedback-based synthetic optimization**
>
>    Prior influence-based methods for **data selection** (e.g., LESS and recent data valuation / active selection works) typically operate as post-hoc, static filters: they score an existing pool of (real or synthetic) examples and select a subset for training, or heuristically modify the synthesized rubrics based on the observed training outcomes. Such approaches are generally inefficient and yield unstable performance. In contrast, our framework:
>
>    - Does not perform a one-shot “influence ranking then filter” step.
>    - Instead, we combine **influence signals**, lightweight format constraints, and safety checks into a verifiable reward, and use this reward to drive **RL over a rubric generator**, which is treated as the policy.
>    - This induces a closed-loop optimization along the entire chain  $\text{seed document} \rightarrow \text{rubric} \rightarrow \text{synthetic QA}$ rather than only at the “final data pool” stage.
>
>    In other words, we move from “influence-guided selection of synthetic data” to “influence-guided generation and shaping of the rubrics that produce synthetic data,” which leads to a qualitatively different control point in the pipeline.
>
> ---
>
> 2. **Conceptual insight: embedding space vs gradient space, and rubrics as strategies**
>
>    Beyond the algorithmic pipeline, we explicitly address a more conceptual question:
>
>    > Why do many synthetic samples or rules that look “semantically reasonable” still fail to improve training?
>
>    To this end, we systematically compare embedding space and **gradient space**:
>
>    - We observe that synthetic and real samples can be very close in embedding space yet exhibit **drastically different influence**, i.e., their contributions to the student’s update direction differ substantially.
>    - Conversely, synthetic sets whose gradients align well with the validation gradient distribution are the ones that actually improve downstream performance.
>
>    This helps explain why many “semantically in-distribution” synthetic examples perform poorly in practice, and supports a more general view:
>
>    > *Rubrics should not merely be hand-crafted prompts based on human intuitions; they can be treated as strategy objects that are quantitatively evaluated and optimized using model gradient feedback.*
>
> Compared to traditional “quality scores” or “difficulty scores” used for heuristic filtering, our formulation upgrades the synthetic pipeline to **model-impact supervision**, where rubrics are optimized to match the effect of real high-impact data, rather than only their surface semantics.

---

> ### Author Response · Authors · 2025-11-21
> **Response to Weakness 2 & Question 1**
>
> > W2: Scalability and Efficiency Unclear. Traditional influence functions are notoriously computationally expensive. The paper claims to deploy an “optimizer-aware estimator” that scales efficiently, but does not provide sufficient empirical evidence or runtime comparisons to support this claim. Since the method is applied per-sample, a clear analysis of runtime overhead and system-level feasibility is crucial to evaluate its practicality for real-world synthetic data pipelines.
>
> We thank the reviewer for raising this important point about scalability and system-level feasibility.
>
> **1. Estimator design and theoretical complexity.**
>
> Our influence estimator is directly built on the optimizer-aware, efficient estimator proposed in LESS [1], which has already been validated on large-scale LM settings. We follow the same two key design choices:
>
>   - Low-rank parameter subspace. We restrict computation to a LoRA parameter subspace instead of the full model parameter space, so that all gradients and updates are computed on a much smaller set of parameters.
>   - Low-dimensional gradient embeddings. We further project per-sample gradients via random projection into a low-dimensional feature space of dimension $d$.
>
>     Let $|D|$ be the number of candidate training samples, $N$ the number of checkpoints used to approximate the training trajectory, and $d$ the projection dimension. Then the asymptotic complexity of our pipeline is:
>
>   - Gradient feature computation (per-sample influence features).
>   For each sample, we compute Adam-style gradients in the LoRA subspace at $N$ checkpoints and project them to dimension $d$. The total cost is $O(|D| \cdot N)$, which is linear in the number of samples and independent of the full model parameter size beyond the fixed LoRA rank.
>
>   - Influence computation.
>   After building the gradient feature bank, we only operate in the low-dimensional feature space, computing similarities between training samples and a small validation set $D_{\text{val}}$. The complexity is $O(|D| \cdot |D_{\text{val}}| \cdot d)$, and does not require any additional forward/backward passes through the model.
>
>     In contrast, traditional second-order influence-function methods rely on explicit (or approximate) Hessian inverses or repeated Hessian–vector products, with at least quadratic dependence on parameter count and large constant factors, which makes them effectively infeasible for modern LLMs. Our estimator is strictly **first-order**, LoRA-restricted, and operates in a compressed feature space, yielding a per-sample cost that scales linearly with $|D|$ and is practical for hundreds of thousands of instructions.
>
> ---
>
> **2. Runtime overhead and system-level feasibility.**
>
> We also report the wall-clock runtime (measured as single H200 GPU-hours) and storage cost of each step in our pipeline in the revised version. Empirically, the influence computation is lightweight:
>
>   - The end-to-end per-sample influence score (including gradient feature computation + similarity in the projected space) takes ≈0.14 seconds per sample on a single H200 GPU under our setup.
>   - For a typical synthetic instruction pool of $\mathcal{O}(10^5)–\mathcal{O}(10^6)$ candidates, this corresponds to a few to tens of GPU-hours of offline preprocessing, which is small compared to the cost of full SFT training on the same model.
>
>     Importantly, this influence estimation is performed once per candidate pool and can be re-used for multiple rounds of rubric optimization and synthetic generation. In our full pipeline, the runtime overhead of influence computation accounts for only a minor fraction of the total time budget.
>
> ---
>
> **3. Clarification on “per-sample” application.**
> While the method produces **per-sample influence scores**, the heavy lifting (model gradients) is amortized via:
>   - shared gradients over **LoRA subspace checkpoints**, and
>   - vectorized projection and similarity computation in a low-dimensional space.
>
>     Thus, “per-sample” here should be understood as a **vectorized batched operation with linear scaling**, rather than independent full-model second-order computations per example.

---

> > ### Author Response · Authors · 2025-11-21
> > **Response to Weakness 3 & Question 2**
> >
> > > W3: Questionable Necessity of the Influence Signal. The paper argues that influence estimation provides a better feedback signal for synthetic data quality, but the empirical justification is not fully convincing. The improvement margins over simpler proxy metrics (e.g., downstream reward, embedding-similarity heuristics, or small proxy validation sets) are modest. Without ablation or sensitivity studies, it remains unclear whether the influence-based signal is truly superior or merely another heuristic choice. A comparison against alternative feedback mechanisms would strengthen the paper’s argument.
> >
> > > Q2. Have the authors compared their influence-based signal with alternative feedback mechanisms?
> >
> > We thank the reviewer for raising this important question about whether the influence-based signal is truly necessary, rather than just another heuristic.
> >
> > **1. Ablation over feedback mechanisms.**
> >
> > In the revised version, we add a dedicated ablation study that directly compares our influence-based feedback against several representative alternatives, while keeping all other components fixed (same teacher, same seeds, same training pipeline). We only replace the feedback used to guide rubric optimization and data selection:
> >
> >   - **QuRating [1]**: A learned quality scorer trained from LLM pairwise judgments along multiple dimensions (style, required expertise, factuality/knowledge content, educational value). We use QuRating-style quality scores as a proxy reward for data selection.
> >   - **FineWeb-Edu proxy [2]**: FineWeb-Edu filters high-quality educational web text and shows strong gains on knowledge- and reasoning-centric benchmarks. Inspired by this, we construct a proxy that measures similarity to the FineWeb-Edu distribution as an “educationality/knowledge density” score.
> >   - **PRRC / Meta-rater [3]**: PRRC evaluates data quality along Professionalism, Readability, Reasoning, Cleanliness. We follow this multi-dimensional scoring approach and treat the combined PRRC score as another proxy reward / proxy-validation signal.
> >   - **Ours (Influence)**: The proposed influence-based signal aligned with the target validation loss.
> >
> >     On six MH benchmarks, the results are:
> >
> >     | Feedback Mechanism | MMLU-Pro | SuperGPQA | HLE    | HealthBench | PubMed | MedQA  |
> >     |--------------------|---------:|----------:|-------:|------------:|-------:|-------:|
> >     | Base Model         | 0.4597   | 0.2806    | 0.1036 | 0.7006      | 0.6590 | 0.5145 |
> >     | QuRating           | 0.4954   | 0.3032    | 0.1118 | 0.7106      | 0.7356 | 0.5534 |
> >     | FineWeb-Edu        | 0.4682   | 0.2892    | 0.1064 | 0.7340      | 0.7044 | 0.5628 |
> >     | PRRC / Meta-rater  | 0.5317   | 0.3472    | 0.0914 | 0.7556      | 0.7568 | 0.5247 |
> >     | Ours (Influence) | 0.5697 | 0.3828  | 0.1081 | 0.7482    | 0.8070 | 0.5875 |
> >
> >   Thus, all three simpler proxies significantly outperform the base model (confirming that “downstream reward / proxy validation” signals are indeed effective), but the influence-based signal still achieves the **best average performance**, with a +3.3 point absolute gain over the strongest proxy (PRRC) and +8.1 over base on this MH suite. The improvements are moderate but **consistent across all six benchmarks**, indicating that influence is not just interchangeable with simpler proxies.
> >
> > ---
> >
> > **2. Why influence is conceptually different from proxy signals.**
> >
> >   We agree that influence is not the only way to obtain useful feedback. However, it is qualitatively different from the proxy mechanisms we tested:
> >
> >   - Proxy signals such as QuRating, FineWeb-Edu similarity, and PRRC scores primarily depend on heuristic properties of the sample itself: e.g., teacher-assigned quality scores, similarity to curated high-quality corpora, or local performance on a small validation set. They measure how good a sample looks according to pre-defined criteria.
> >   - In contrast, the influence-based signal is derived from an optimization perspective: it approximates “If we slightly upweight this sample under the current parameters and training trajectory, how much will the target validation loss change?”
> >
> >   This directly ties the feedback to the actual training dynamics and the specific target validation set we care about, rather than to purely surface-level or local proxies.
> >     In other words, influence is designed to measure each sample’s marginal contribution to the final loss on the target task, whereas proxy rewards measure various notions of instantaneous or heuristic quality. Our ablation indicates that this alignment with the true target objective yields consistent additional gains on top of strong proxies.
> >
> > [1] QuRating: Selecting High-Quality Data for Training Language Models
> >
> > [2] The FineWeb Datasets: Decanting the Web for the Finest Text Data at Scale
> >
> > [3] Meta-rater: A Multi-dimensional Data Selection Method for Pre-training Language Models

---

> ### Author Response · Authors · 2025-11-26
> **Supplementary experiment: Quantifying computational efficiency (Part 1)**
>
> In response to your suggestion to better quantify compute efficiency, we add additional analysis and experiments. Following the FLOPs estimation methodology in [X], we estimate the FLOPs of Montessori-Instruct and OptimSyn under a unified setting. Montessori-Instruct treats samples that improve vs. degrade performance as positive/negative pairs, and then applies DPO to train the teacher model on these pairs.
>
> ---
>
> ### Unified setting and FLOPs model
>
> - Student / target / prompter: 8B params, $N_8 = 8\times 10^9$
> - Teacher: 30B params, $N_{30} = 3\times 10^{10}$
> - Data: 26,600 samples, 125.7 tokens each  $\Rightarrow D = 26{,}600 \times 125.7 \approx 3.34\times 10^6$ tokens
>
> FLOPs approximation:
>
> - Forward only: $\text{FLOPs} \approx 2ND$
> - Training (forward + backward): $\text{FLOPs} \approx 6ND$
>
> Single SFT of 8B on the full data:
>
> - $F_{\text{SFT}}^{8\text{B}} = 6 N_8 D \approx 1.60\times 10^{17}$ FLOPs
>
> ---
>
> ### **1. Montessori-Instruct** (8B student + 30B teacher)
>
> Assumptions: warm-up on $0.1D$, local influence over $D$, DPO on $D$, teacher generates 4 rounds on $D$, then final SFT on $D$.
>
> - Warm-up (8B, $0.1D$):  $F_{\text{warm}}^{M} = 6 N_8 (0.1D) \approx 1.60\times 10^{16}$ FLOPs
>
> - Local influence (8B, $\approx 2$ passes on $D$):  $F_{\text{influence}}^{M} \approx 12 N_8 D \approx 1.28\times 10^{18}$ FLOPs
>
> - DPO (30B, $\approx 2\times$ SFT on $D$):  $F_{\text{DPO}}^{M} \approx 12 N_{30} D \approx 1.20\times 10^{18}$ FLOPs
>
> - Teacher generation (30B, 4 rounds on $D$):  $F_{\text{teacher-gen}}^{M} = 8 N_{30} D \approx 8.02\times 10^{17}$ FLOPs
>
> - Final SFT (8B on $D$):   $F_{\text{final-SFT}}^{M} \approx 1.60\times 10^{17}$ FLOPs
>
> Total Montessori-Instruct (one iteration):
>
> - $F_{\text{total}}^{M} = F_{\text{warm}}^{M} + F_{\text{influence}}^{M} + F_{\text{DPO}}^{M} + F_{\text{teacher-gen}}^{M} + F_{\text{final-SFT}}^{M} \approx 3.50\times 10^{18}$ FLOPs
>
> ---
>
> ### **2. OptimSyn**
>
> We assume RL draws $n=5$ QA per sample:
>
> - QA tokens: $D_{\text{RL,QA}} = 5D$
> - Rubric tokens: $D_{\text{RL,rubric}} \approx \frac{1}{3} D_{\text{RL,QA}} = \frac{5}{3}D$
>
> Teacher first generates base QA on $D$, then answers for $5D$ QA tokens.
>
> 8B-side:
>
> - Warm-up ($0.1D$):  $F_{\text{warm}}^{O} = 6 N_8 (0.1D) \approx 1.60\times 10^{16}$ FLOPs
>
> - Influence caching (overhead ≈ 10% of warm-up):  $F_{\text{cache}}^{O} \approx 0.1 F_{\text{warm}}^{O} \approx 1.60\times 10^{15}$ FLOPs
>
> - Prompter GRPO on rubrics ($D_{\text{RL,rubric}}$):  $F_{\text{prompter-GRPO}}^{O} = 6 N_8 D_{\text{RL,rubric}} = 10 N_8 D \approx 2.67\times 10^{17}$ FLOPs
>
> - Target influence on QA ($D_{\text{RL,QA}} = 5D$):  $F_{\text{target-infl}}^{O} = 30 N_8 D \approx 8.02\times 10^{17}$ FLOPs
>
> - Final SFT ($D$):  $F_{\text{final-SFT}}^{O} \approx 1.60\times 10^{17}$ FLOPs
>
> Total 8B-side:
>
> - $F_{\text{8B-total}}^{O} = F_{\text{warm}}^{O} + F_{\text{cache}}^{O} + F_{\text{prompter-GRPO}}^{O} + F_{\text{target-infl}}^{O} + F_{\text{final-SFT}}^{O} \approx 1.25\times 10^{18}$ FLOPs
>
> 30B teacher-side:
>
> - Base QA generation on $D$:  $F_{\text{teacher-base}}^{O} = 2 N_{30} D \approx 2.01\times 10^{17}$ FLOPs
>
> - RL answer generation on $5D$:   $F_{\text{teacher-RL}}^{O} = 10 N_{30} D \approx 1.00\times 10^{18}$ FLOPs
>
> Total teacher-side:
>
> - $F_{\text{teacher-total}}^{O} = F_{\text{teacher-base}}^{O} + F_{\text{teacher-RL}}^{O} \approx 1.20\times 10^{18}$ FLOPs
>
> Overall OptimSyn:
>
> - $F_{\text{total}}^{O} = F_{\text{8B-total}}^{O} + F_{\text{teacher-total}}^{O} \approx 2.45\times 10^{18}$ FLOPs
>
> ---
>
> ### 3. Summary
>
> Under the same model (8B student + 30B teacher) and data scale (26.6k samples, 125.7 tokens):
>
> - Montessori-Instruct (1×): $F_{\text{total}}^{M} \approx 3.50\times 10^{18}$ FLOPs
> - OptimSyn: $F_{\text{total}}^{O} \approx 2.45\times 10^{18}$ FLOPs
>
> Thus, our method and Montessori-Instruct are in the same compute regime (order $3.5\times 10^{18}$ FLOPs per full run), and **OptimSyn is cheaper (about 42% in this setting)**, indicating that our approach is compute-comparable and practically feasible relative to existing methods. We also compare the downstream performance of models trained on synthetic data generated by OptimSyn and Montessori-Instruct in Table 1.
>
> | Method              | MMLU pro | Super GPQA | HLE    | Health Bench | PubMed | MedQA  |
> |---------------------|----------|------------|--------|--------------|--------|--------|
> | Base                | 0.4597   | 0.2806     | 0.1036 | 0.7006       | 0.659  | 0.5145 |
> | Monressor-Instruct  | 0.4672   | 0.2728     | 0.0946 | 0.6453       | 0.7027 | 0.5204 |
> | Monressor (inter 2) | 0.5028   | 0.3105     | 0.1047 | 0.6893       | 0.7458 | 0.5469 |
> | Ours                | 0.5369   | 0.3549     | 0.1096 | 0.7062       | 0.7859 | 0.5206 |
>
>
> [1] Hoffmann J, et al. Training compute-optimal large language models[J]. NIPS 2022.
>
> [2] Li X, Yu Z, Xiong C. Montessori-instruct: Generate influential training data tailored for student learning[J]. ICLR 2025.

---

> > ### Author Response · Authors · 2025-11-26
> > **Supplementary experiment: Quantifying computational efficiency (Part 2)**
> >
> > We further compare the training efficiency of our method against baseline datasets. The dataset statistics are summarized in Table 2. Since all models are trained with the same SFT framework and under identical settings, the required training FLOPs can be approximated by the total number of tokens in each dataset. As shown in Tables 3, under comparable compute budgets, our OptimSyn-based model consistently achieves the best performance. We have updated the PDF accordingly, and the corresponding visualizations are provided in Appendix E.7 (Training Efficiency Comparison).
> >
> > **Table 2 Data Statistic**
> > | Dataset               | Token (avg) | Num.  |
> > |-----------------------|------------:|------:|
> > | ChatDoctor            | 246.3       | 100K  |
> > | MedQA                 | 15.7        | 12.7k |
> > | Medical-o1            | 557.9       | 19.7k |
> > | Medical-R1-Distill    | 579.4       | 22k   |
> > | ReasonMed             | 645.4       | 370k  |
> > | OptimSyn (ours)       | 125.7       | 26.6k |

---

> > > ### Author Response · Authors · 2025-11-26
> > > **Supplementary experiment: Quantifying computational efficiency (Part 3)**
> > >
> > > **Table3 Comparison of SFT results under the same computational budget.**
> > >
> > > | Benchmark   | Dataset             | 0     | $1.20\times 10^{16}$ | $6\times 10^{16}$ | $2.40\times 10^{17}$ | $3.60\times 10^{17}$ | $4.80\times 10^{17}$ | $6.00\times 10^{17}$ | $1.20\times 10^{18}$ | $1.20\times 10^{19}$ |
> > > |------------|---------------------|------:|---------------------:|------------------:|----------------------:|----------------------:|----------------------:|----------------------:|----------------------:|----------------------:|
> > > | MMLU pro   | ChatDoctor          | 45.97 | 47.32                | 46.28             | 35.07                 | 30.24                 | 29.94                 | 30.48                 | 16.48                 | -                    |
> > > |  | MedQA               | 45.97 | 13.45                | -                 | -                     | -                     | -                     | -                     | -                     | -                    |
> > > |   | Medical-o1          | 45.97 | 49.36                | 53.96             | 56.48                 | 59.31                 | 60.83                 | 60.51                 | -                     | -                    |
> > > |    | Medical-R1-Distill  | 45.97 | 48.63                | 50.84             | 52.10                 | 54.96                 | 56.57                 | 57.33                 | -                     | -                    |
> > > |    | ReasonMed           | 45.97 | 46.96                | 48.34             | 50.93                 | 52.49                 | 53.84                 | 54.94                 | 53.92                 | 55.00                |
> > > |    | OptimSyn            | 45.97 | 52.46                | 54.96             | 56.97                 | -                     | -                     | -                     | -                     | -                    |
> > > | Super GPQA | ChatDoctor          | 28.06 | 28.84                | 27.65             | 26.54                 | 27.32                 | 25.01                 | 23.38                 | 21.22                 | -                    |
> > > |  | MedQA               | 28.06 | 34.61                | -                 | -                     | -                     | -                     | -                     | -                     | -                    |
> > > |  | Medical-o1          | 28.06 | 26.53                | 27.61             | 28.94                 | 25.90                 | 27.84                 | 27.26                 | -                     | -                    |
> > > |  | Medical-R1-Distill  | 28.06 | 30.68                | 32.86             | 33.57                 | 35.24                 | 34.86                 | 35.28                 | -                     | -                    |
> > > |  | ReasonMed           | 28.06 | 28.86                | 27.94             | 27.56                 | 28.03                 | 29.31                 | 27.24                 | 27.59                 | 28.01                |
> > > |  | OptimSyn            | 28.06 | 30.58                | 35.74             | 38.28                 | -                     | -                     | -                     | -                     | -                    |
> > > | PubMed     | ChatDoctor          | 65.90 | 69.43                | 73.28             | 75.94                 | 78.52                 | 80.77                 | 84.36                 | 85.40                 | -                    |
> > > |     | MedQA               | 65.90 | 71.70                | -                 | -                     | -                     | -                     | -                     | -                     | -                    |
> > > |     | Medical-o1          | 65.90 | 70.62                | 71.36             | 73.58                 | 76.57                 | 73.83                 | 73.90                 | -                     | -                    |
> > > |     | Medical-R1-Distill  | 65.90 | 69.51                | 72.36             | 74.02                 | 74.58                 | 75.24                 | 73.40                 | -                     | -                    |
> > > |     | ReasonMed           | 65.90 | 70.27                | 73.73             | 76.52                 | 78.92                 | 78.64                 | 79.05                 | 80.12                 | 79.80                |
> > > |     | OptimSyn            | 65.90 | 72.86                | 78.90             | 80.70                 | -                     | -                     | -                     | -                     | -                    |
> > >
> > > We hope the above supplementary experiments can further address your concerns. If there are any other questions, we will continue to refine our work. We look forward to your reply. Best wish for you.

---

> > > > ### Comment · Reviewer_WCjb · 2025-11-28
> > > > **Thank you for the detailed rebuttal and question about high-level understanding**
> > > >
> > > > Thank you for the very detailed rebuttal and all the additional analyses. I really appreciate the effort. My remaining concerns are conceptual rather than experimental, so please don’t spend more time running new experiments. I like the closed-loop, feedback-based synthetic optimization view, but the choice of influence functions still feels somewhat heuristic, almost like a training-free GRPO with another reward. Since you emphasize IF as “optimization-aware,” I would really value a clearer argument for why this optimization-space signal is fundamentally preferable to other plausible signals, and why it is worth going into LoRA parameter space instead of staying in other space. In short, my concern is that the choice of both the space and the signal appears heuristic rather than principled or evidence-driven. If you could clarify this conceptual story, I would really appreciate it.

---

> > > > > ### Author Response · Authors · 2025-11-28
> > > > > **Response to the question about high-level understanding (Part 1)**
> > > > >
> > > > > Thank you very much for the detailed comments and for pushing us to clarify the conceptual story.
> > > > >
> > > > > ### **1. Why do we choose influence functions as the reward signal?**
> > > > >
> > > > > Our starting point is that, in the synthetic data setting, the most fundamental measure of data quality is its **actual improvement on downstream task performance after training**. However, the common practice today is often a brittle heuristic loop:
> > > > >
> > > > > > write a rubric → synthesize data → train a model → inspect downstream metrics → guess how to tweak the rubric.
> > > > >
> > > > > This loop has two main issues:
> > > > > (i) the feedback on how much a particular rubric / data subset truly contributes to downstream performance is delayed and coarse; and
> > > > > (ii) the feedback signal itself is not grounded in the optimization process, and thus is not principledly tied to the training dynamics of the target model.
> > > > >
> > > > > Influence functions are designed precisely to fill this gap. In our setting, the Adam-compatible IF estimator approximates
> > > > > $\frac{\partial L_{\text{val}}}{\partial \alpha_z}$,
> > > > > i.e., **how the validation loss** $L_{\text{val}}$ **would change if we slightly upweight a particular training example** $z$ **from weight** $1$ to $1 + \alpha$. This quantity is derived directly from the training objective and a first-order Taylor expansion along the actual optimization trajectory, with the specific optimizer (e.g., Adam’s momentum and second moments) encoded in the estimator.
> > > > >
> > > > > Therefore:
> > > > >
> > > > > - it is not an ad-hoc heuristic score;
> > > > > - once we decide we want to approximate “the marginal effect of a single example on validation loss”, it is essentially the natural first-order approximation.
> > > > >
> > > > > From this perspective, IF is not “just another reward design”, but a feedback signal that is tightly coupled with the training objective and the optimization process.
> > > > >
> > > > > Below we compare IF with two major families of model-based signals.
> > > > >
> > > > > ---
> > > > >
> > > > > **(1) Comparison with content-driven heuristic signals**
> > > > >
> > > > > This family includes, for example: semantic / embedding similarity to high-quality corpora, LLM-as-judge quality scores (QuRating-like), “educational density” (FineWeb-Edu style), and multi-dimensional quality scores (Meta-rater / PRRC style).
> > > > >
> > > > > They all focus on content properties of an example (professionalism, textbook-likeness, similarity to a reference set), but do not model how this example will update the target model’s parameters. In this sense they are largely model-agnostic, only weakly connected to the specific model and optimizer we actually train.
> > > > >
> > > > > By contrast, IF explicitly depends on the current model and its optimization trajectory, and directly answers: *“If the training process pays more attention to this example, will the validation loss improve or worsen?”* To the best of our knowledge, IF is one of the very few feedback mechanisms defined directly in terms of the change in validation loss induced by an update, rather than static content features.
> > > > >
> > > > > ---
> > > > >
> > > > > **(2) Comparison with uncertainty-based model feedback**
> > > > >
> > > > > Uncertainty-based signals, derived from $p_\theta(y \mid x)$ (Least Confidence, Margin, Entropy), are classical in active learning and typically stronger than pure content heuristics. However, even as model-based signals, they differ fundamentally from IF:
> > > > >
> > > > > - They depend only on the output distribution, answering *“How unsure is the model about this example now?”*,
> > > > > - but not *“If we train on this example, will the resulting parameter update help the validation task?”*.
> > > > > - With a strong teacher, highly uncertain examples often correspond to ambiguous labels, low-quality prompts, or extreme long-tail cases, which are not guaranteed to help downstream training and can even hurt.
> > > > >
> > > > > IF instead is defined via gradient alignment. Let
> > > > > $g(z_{\text{train}}) = \nabla_\theta \ell(z_{\text{train}})$ and
> > > > > $g(z_{\text{val}}) = \nabla_\theta \ell(z_{\text{val}})$
> > > > > be gradients w.r.t. parameters $\theta$. The per-step influence is
> > > > > $\text{Influence}(z_{\text{train}}, z_{\text{val}}) \propto \langle g(z_{\text{train}}), g(z_{\text{val}}) \rangle$.
> > > > >
> > > > > A training example is useful if its gradient pushes parameters in a direction aligned with the validation gradient:
> > > > >
> > > > > - Some highly uncertain examples can be anti-aligned, yielding negative IF and should be downweighted.
> > > > > - Some “easy” examples (low entropy) can still reduce validation loss and thus receive high IF, even though uncertainty-based methods would ignore them.
> > > > >
> > > > > In short, uncertainty signals ask *“Does the model currently understand this example?”*, whereas IF asks *“If we train on this example, will it actually improve the downstream task?”*. For synthetic data optimization, we care about the latter, and IF is one of the few signals that directly links model feedback to the effect of future optimization, not just current prediction difficulty.
> > > > >
> > > > > ---
> > > > >
> > > > > **(3) Empirical evidence**
> > > > >
> > > > > Our empirical experiments also show that IF-based rewards consistently outperform these alternatives in both performance and stability.

---

> ### Author Response · Authors · 2025-11-28
> **Response to the question about high-level understanding (Part 2)**
>
> ### **2. Why operate in parameter / LoRA space?**
>
> By definition, **influence is a notion about parameter updates**. For a training example $z_{\text{train}}$ and a validation example $z_{\text{val}}$, the per-step influence used in our method can be written (up to a constant factor) as:
>
> $\text{Influence}(z_{\text{train}}, z_{\text{val}}) \propto
> \langle g(z_{\text{train}}), g(z_{\text{val}}) \rangle,$
>
> where $g(\cdot)$ denotes the gradient of the training objective with respect to the model parameters. Intuitively, a training example is *useful* if and only if its gradient pushes the parameters in a direction that is aligned with the gradient induced by the validation task. This is inherently a directional alignment notion in parameter space: if we were to replace gradients with sentence embeddings or token-level representations, we would be measuring *semantic similarity* rather than *alignment of update directions*.
>
> We therefore choose to compute these gradients in the **LoRA parameter subspace**, for both conceptual and practical reasons:
>
> - **Conceptually**, downstream SFT in our setup is implemented via LoRA / adapters: the backbone parameters are frozen, and only the low-rank LoRA adapters at each layer are updated. Thus, computing influence with respect to these trainable LoRA parameters directly answers the question *“how would this example affect the actual task-specific adaptation we perform?”*, rather than *“what would happen if we perturbed the entire frozen backbone?”*.
>
> - **Practically**, the LoRA subspace has a much lower dimensionality than the full parameter space, which implies that:
>   - per-example gradients are significantly cheaper to compute and store;
>   - inner-product estimates between gradients have lower variance, yielding more stable influence scores that are less dominated by high-dimensional noise.
>
>
> Since we noticed that the system does not allow changing the score directly, if you are open to increasing your score, we would kindly ask you to mention this explicitly in your response. In addition, if you have any further questions or would like to see additional experiments, we would be very happy to hear from you and will continue to improve and extend the work accordingly.

---

### Official Review · Reviewer_tUAj · 2025-11-03

**Soundness:** 3
**Presentation:** 3
**Contribution:** 3
**Rating:** 6
**Confidence:** 4

**Summary:**

Previous studies on synthetic data generation typically rely on expert-designed, domain-specific rubrics, which often suffer from limited transferability and scalability. This paper proposes a novel paradigm: instead of handcrafting rubrics, it advocates evaluating synthetic data quality based on its actual training utility on the target model. This feedback is then used to iteratively refine the rubrics themselves, forming a closed loop between data synthesis and downstream performance.

The resulting QA pairs are optimized to maximize estimated downstream benefit under these learned rubrics. Experiments in two low-resource domains—Humanities & Social Sciences (HSS) and Medical & Health (MH)—demonstrate that the learned rubrics are both model- and task-conditioned, reducing reliance on domain expertise.

**Strengths:**

1.	The key insight—replacing heuristic rubric design with a model-impact-driven objective—is interesting. Training a rubric generator to maximize downstream utility offers a principled and adaptive alternative to manual engineering.
2.	Evaluation in HSS and MH domains, where high-quality SFT data is scarce, showcases the method’s practical value and generalizability beyond standard benchmarks.

**Weaknesses:**

1.	How do the number and length of rubrics evolve throughout reinforcement learning? Could the observed improvements be primarily driven by the generator's capacity or the diversity of synthesized QA pairs, rather than the rubrics themselves? What mechanism controls the ratio between rubric guidance and QA pairs? Additionally, have you analyzed the diversity of the generated data and its correlation with specific rubrics?
2.	Could the authors provide more details about the lightweight verification rules (e.g., formatting correctness, non-triviality, safety filters)?
3.	Are there notable differences in data statistics (e.g., size, average length, complexity) between prior datasets and the proposed one? Such differences may confound the observed gains; a comparative analysis would strengthen the claims.
4.	In the 14B model results, the performance gain in the MH domain appears less pronounced than in HSS. Have the authors explored whether increasing the volume of synthetic data improves outcomes in MH, suggesting a data-hungry regime?

**Questions:**

please refer to "Weaknesses"

---

> ### Author Response · Authors · 2025-11-21
> **Response to Weakness 1.1**
>
> In response to your comments, we have also made corresponding additions and revisions in the submitted PDF version of the paper. Due to space constraints, the new content is currently included in the appendix, but we will integrate these analyses into the main text in the final open-source version of the paper. Specifically, we highlight in `red` all additions made based on your feedback: (i) a more detailed analysis of the baseline data characteristics in Appendix D.2 BASELINE ANALYSIS, and (ii) a more systematic study and discussion of the impact of data scale in Appendix E.3 ANALYSIS.
>
> ---
>
> > W1.1: How do the number and length of rubrics evolve throughout reinforcement learning? Additionally, have you analyzed the diversity of the generated data and its correlation with specific rubrics?
>
> We thank the reviewer for this question about the evolution of rubrics and their relationship to data diversity.
>
> **1.Number and length of rubrics during RL.**
>
>   - In our RL setup, the number of rubrics per seed document is kept constant: for each seed we always sample a fixed rollout group of G rubrics (Algorithm 1, “rollout group size G”), so RL changes the content of rubrics rather than their count.
>   - To characterize how rubrics evolve, we tracked their average token length and lexical diversity at multiple RL checkpoints (steps 8–88). As shown in Table R1 (to be added to the appendix), the average rubric length steadily decreases from 128.8±33.0 tokens at step 8 to 85.7±24.6 tokens at step 88 (≈33% reduction), while rubric lexical diversity (MTLD) increases from 67.0 to 95.9 and HD-D rises from 0.80 to 0.90. This indicates that RL makes rubrics more concise but more informative, rather than simply “longer prompts”.
>
> **2.Diversity of generated QA data and its correlation with rubrics.**
>
>   - For the synthesized QA pairs, we measured both length and lexical diversity along the RL trajectory. Using MTLD and HD-D (standard length-robust diversity measures), we observe that the average QA length increases from 133.0 to 195.1 tokens (+46.6%), and QA lexical diversity also increases (MTLD: 125.7 → 135.7; HD-D remains high and stable). This suggests that the generator produces richer and more detailed QA pairs as rubrics improve.
>   - To directly link rubrics to data diversity, we group QA pairs by the RL checkpoint of the rubrics that generated them and compute correlations between rubric diversity and QA diversity across these checkpoints. We obtain a strong positive Pearson correlation between rubric MTLD and QA MTLD (r ≈ 0.77), while rubric length is negatively correlated with QA length (r ≈ −0.97). In other words, shorter but lexically richer rubrics systematically lead to longer and more diverse QA pairs.
>   - Qualitatively, Fig. 6 in the paper already visualizes this evolution: t-SNE embeddings show that post-RL rubrics cover a broader, more structured region in the generator space, and the word clouds shift from generic imperatives (e.g., focus, align, short) to domain- and quality-oriented criteria (e.g., critical, clarity, completeness, logical soundness). These more specific rubrics better constrain the generator and are consistent with the observed gains in QA diversity and utility.
>
> | Metric                | Step 8   | Step 16  | Step 24  | Step 32  | Step 40  | Step 48  | Step 56  | Step 72  | Step 80  | Step 88  |
> |-----------------------|----------|----------|----------|----------|----------|----------|----------|----------|----------|----------|
> | length of QA          | 133.0413 | 159.9187 | 153.8003 | 164.8577 | 175.1452 | 167.2326 | 190.9512 | 184.7814 | 196.4916 | 195.0672 |
> | std (QA)              | 42.7047  | 44.6677  | 40.7035  | 55.6776  | 44.6744  | 49.6737  | 45.6976  | 48.6721  | 49.6966  | 48.5754  |
> | Diversity of QA       | 125.698  | 129.8958 | 126.291  | 126.5873 | 128.2183 | 129.6928 | 132.2037 | 135.7173 | 133.8239 | 130.8364 |
> | HD-D (QA)             | 0.8943   | 0.8942   | 0.9044   | 0.9343   | 0.8842   | 0.8942   | 0.9143   | 0.9242   | 0.9241   | 0.9273   |
> | length of Rubrics     | 128.8002 | 116.9179 | 115.0145 | 104.0702 | 93.1969  | 98.9872  | 88.9152  | 92.973   | 85.046   | 85.6924  |
> | std (Rubrics)         | 32.9975  | 32.8787  | 30.9569  | 29.9601  | 29.1663  | 33.0411  | 25.9028  | 27.9039  | 24.0097  | 24.6389  |
> | Diversity of Rubrics  | 66.9748  | 67.1874  | 78.3116  | 72.9991  | 84.3239  | 92.1911  | 89.2379  | 97.3467  | 97.3981  | 95.8701  |
> | HD-D (Rubrics)        | 0.795    | 0.7947   | 0.8049   | 0.8395   | 0.9152   | 0.8748   | 0.8947   | 0.8944   | 0.9052   | 0.9      |

---

> > ### Author Response · Authors · 2025-11-21
> > **Response to Weakness 1.2 & Weakness 1.3**
> >
> > > W1.2: Could the observed improvements be primarily driven by the generator capacity or the diversity of synthesized QA pairs, rather than the rubrics themselves?
> >
> > We appreciate the reviewer’s question about potential confounding factors.
> >
> > 1. **Controlling for generator capacity and training setup**
> >
> >    In our main comparison between “Before RL” and **“Ours”**, we use the same teacher/generator, the same set of seed documents, and the same training pipeline and hyperparameters. The only difference is whether the rubric generator has been optimized by RL. Under this controlled setting, any observed changes in QA quality or diversity are necessarily induced by the rubrics (i.e., by replacing the pre-RL rubrics with RL-optimized rubrics), rather than by changes in generator capacity or other components. This indicates that the RL-trained rubric generator provides more effective guidance to the generator for synthesizing data.
> >
> > 2. **Effect of generator capacity**
> >
> >    We further explicitly examine the impact of the teacher/generator capacity in Sec. 3.3. Concretely, we replace the generator with **Qwen3-235B**, **GPT-4.1**, and **Gemini-2.5-Pro**, while keeping all other settings (rubric learning procedure, seed documents, training recipe) unchanged. Across all these generators, models trained on data produced by our rubric-optimized pipeline consistently outperform their respective baselines trained on data without RL-optimized rubrics. This cross-generator consistency demonstrates that:
> >
> >    - the gains are not tied to a particular generator or its raw capacity; and
> >    - the learned rubrics contribute robustly on top of strong generators.
> >
> > 3. **Role of QA diversity vs. rubrics**
> >
> >    We agree that improved QA diversity can be beneficial for downstream performance. In our framework, however, such diversity changes are not an independent variable we tune directly; instead, they **arise as a consequence of improving the rubrics**, under a fixed generator and training setup. Thus, the increased diversity and quality of synthesized QA pairs should be viewed as an effect of better rubrics, rather than an alternative explanation that competes with them.
> >
> >
> > > W1.3: What mechanism controls the ratio between rubric guidance and QA pairs?
> >
> > Thank you for raising this question about how rubric guidance is integrated with QA generation.
> >
> > In our framework, rubrics are never treated as a separate training signal that is mixed with QA supervision via a tunable ratio. Instead, rubrics serve purely as conditional inputs to the teacher/generator:
> >
> > - For each seed document $S$, the policy $\pi$ (rubric generator) produces a rubric $B$.
> > - The teacher then generates exactly one QA pair $(Q, A)$ conditioned on $(S, B)$.
> >
> > Thus, there is a fixed 1:1 correspondence between rubrics and QA pairs in our data-generating process: each rubric deterministically guides the generation of a single QA pair. There is no explicit mixing coefficient or loss-weight hyperparameter that trades off “rubric guidance” against “QA pairs”; QA pairs are the only objects used for supervised fine-tuning of the student model, and rubrics influence the student only indirectly through the QA they induce.
> >
> > During the RL phase for the rubric generator, we use a rollout group of size $G$ to explore multiple rubrics per seed:
> >
> > - For each seed $S$, we sample $G$ rubrics $\{B_1, \dots, B_G\}$ from the policy $\pi$.
> > - Each $B_i$ produces one QA pair $(Q_i, A_i)$ and one scalar reward $r_i$.
> > - These $G$ trajectories are then used to update $\pi$ via policy-gradient RL.
> >
> > Here, $G$ controls the **number of rubric–QA–reward trajectories per seed**, i.e., the exploration breadth in RL, but it does not control a ratio between two different types of supervision. The “ratio between rubric guidance and QA pairs” is effectively fixed by design to 1:1 at the data-generation level.

---

> ### Author Response · Authors · 2025-11-21
> **Response to Weakness 2**
>
> > W2: Could the authors provide more details about the lightweight verification rules (e.g., formatting correctness, non-triviality, safety filters)?
>
> We observe that these lightweight verification rules evolve over the course of training. In the early stages, the model focuses primarily on simple surface-level properties such as formatting consistency (e.g., presence of a clear structure and proper sections), reasonable response length, and basic completeness. As training progresses, domain relevance is invoked much more frequently, and the rules begin to incorporate constraints related to question difficulty, for example requiring that the answer to the question cannot be directly found in the original passage but must be derived through non-trivial reasoning or synthesis. We also find that the rubrics themselves gradually shift from broad, mixed criteria to more detailed and document-specific ones that are tightly aligned with the underlying task and benchmark. To make this process more concrete, we present several examples of rubrics attached to high–influence-score training samples, which qualitatively illustrate how the verification rules become more precise and better tailored to the target domain over time.
>
> ---
>
> ### **Case 1**
>
> **Influence Score**: 0.2827
>
> **Prompt-related**
> "clarity and unambiguity": The question must be unambiguous and directly tied to the document's content (e.g., pathophysiology, diagnostic methods, or treatment options) to ensure the generator focuses on factual extraction rather than speculation.
> "format appropriateness": The question should align with medical terminology and structure (e.g., “What diagnostic imaging modalities are used…?”) to mirror clinical question formats encountered in real-world scenarios.
> "instructional_precision": Specify whether answers should include numerical data (e.g., incidence rates) or procedural details (e.g., surgical techniques) to align with the document's granular content.
>
> **Response-related**
> "factual_accuracy": Answers must strictly adhere to the document's data (e.g., incidence rates, specific surgical procedures) to avoid introducing external knowledge or errors.
> "completeness": Ensure answers include all critical components (e.g., both diagnostic imaging methods mentioned in the document, such as X-ray and CT/MRI) to reflect thorough understanding.
> "information_density": Answers should balance conciseness with detail (e.g., specifying “duodeno-doudenostomy” vs. generalizing to “bypass surgery”) to mirror the precision required in clinical settings.
>
> ---
> ### **Case 2**
>
> **Influence Score**: 0.7276
>
> **Prompt-related**
>
> "Domain relevance": Questions must directly address substance-related disorders, focusing on definitions, classifications (e.g., DSM-5 changes), symptoms, complications, or related concepts (e.g., dual diagnosis) to ensure alignment with the document's core content.
> "Clarity and unambiguity": Prompts should avoid vague phrasing and explicitly specify the scope (e.g., 'Explain the DSM-5 classification of substance use disorders') to prevent misinterpretation and ensure actionable responses.
> "Format appropriateness": Questions must adhere to the selected type (e.g., short answer for concise explanations, multiple-choice for distinguishing between related terms like 'intoxication' vs. 'withdrawal').
>
> **Response-related**
>
> "Factual accuracy": Answers must strictly reflect the document's content (e.g., citing DSM-5's integration of substance abuse/dependence, or listing specific complications like 'alcoholic cardiomyopathy') to ensure medical precision.
> "Information density": Responses should balance conciseness with completeness, avoiding unnecessary details while covering all critical points (e.g., explaining 'dual diagnosis' includes both substance use and psychiatric conditions).
> "Logical soundness": Answers must demonstrate causal or structural reasoning (e.g., linking 'brain cravings' to 'neurobiological mechanisms' as described in the document) to assess analytical comprehension

---

> > ### Author Response · Authors · 2025-11-21
> > **Response to Weakness 3**
> >
> > > W3: Are there notable differences in data statistics (e.g., size, average length, complexity) between prior datasets and the proposed one? Such differences may confound the observed gains; a comparative analysis would strengthen the claims.
> >
> > We thank the reviewer for pointing out the potential impact of basic data statistics on the comparison.
> >
> > **1. Comparative statistics with prior SFT datasets.**
> >
> > In the revised version, we provide a systematic comparison between our proposed dataset OptimSyn and several widely used SFT corpora, including LongWriter, WildChat, Condor-20k, Cosmopedia, OpenHermes 2.5, and SynthQuestions. For each dataset, we report: total number of samples,  average token length and standard deviation,  and lexical diversity, measured by **MTLD** and **HD-D**.
> >
> > | Dataset          | #Samples  | Avg. tokens |   Std |  MTLD  |  HD-D  |
> > |------------------|----------:|------------:|------:|-------:|-------:|
> > | LongWriter       | 6,000     | 1823.45     | 452.20|  80.63 | 0.8954 |
> > | WildChat         | 529,428   | 289.59      | 344.77|  52.70 | 0.9188 |
> > | Condor-20k       | 20,000    | 428.79      |  35.70| 101.48 | 0.8650 |
> > | Cosmopedia       | 50,000    | 773.85      |  44.94| 111.52 | 0.8843 |
> > | OpenHermes 2.5   | 1,001,551 | 236.14      | 249.04| 106.43 | 0.8887 |
> > | SynthQuestions   | 2,500     | 634.60      |  17.88| 137.02 | 0.8584 |
> > | OptimSyn (ours)  | 25,875    | 196.49      |  34.70| 133.82 | 0.9241 |
> >
> >
> > These numbers show that existing SFT datasets already span a very wide range in terms of size and average length (from 2.5K to over 1M samples, and from ~200 to ~1800 tokens on average), as well as lexical diversity. OptimSyn’s size, average length, and diversity all lie well within this existing range; it does not sit at an extreme “larger/longer/richer” corner. Conceptually, OptimSyn is better viewed as a more efficient re-organization and selection of data within the existing statistical envelope, rather than as “just more data” or “much longer texts”.
> >
> > ---
> >
> > **2. Do basic statistics explain the gains?**
> >
> > To directly address the concern about confounding, we correlate these statistics with downstream performance in Table 1 (same target model and training recipe across datasets). We do not observe a monotonic trend such as “larger dataset” or “longer outputs” systematically yielding better performance. In particular:
> >   - Some datasets with larger size or longer average outputs do not achieve the best scores.
> >   - Several moderate-sized datasets with higher-quality supervision (including OptimSyn) show **more stable and stronger performance**.
> >
> >  This analysis suggests that simple data statistics (size, length, lexical diversity) alone cannot account for the improvements brought by our method. Instead, the gains are better explained by the quality and structure of the synthesized data induced by rubric-guided optimization, rather than by raw scale or superficial complexity.

---

> ### Author Response · Authors · 2025-11-21
> **Response to Weakness 4**
>
> > W4: In the 14B model results, the performance gain in the MH domain appears less pronounced than in HSS. Have the authors explored whether increasing the volume of synthetic data improves outcomes in MH, suggesting a data-hungry regime?
>
> We appreciate the reviewer’s careful observation regarding the MH-domain results for the 14B model.
>
> **1. Scaling MH synthetic data: evidence for a data-hungry regime.**
>
> Following the reviewer’s suggestion, we explicitly conducted **data-scaling experiments in the MH domain**, focusing on Qwen3-14B and, for comparison, Qwen3-8B.
>
> | Number        | MMLU pro | Super GPQA |   HLE  | Health Bench | PubMed | MedQA  |
> |--------------|----------|------------|--------|--------------|--------|--------|
> | Qwen3-8B-Base| 0.4597   | 0.2806     | 0.1036 | 0.7006       | 0.6590 | 0.5145 |
> | 5000         | 0.4759   | 0.2654     | 0.1098 | 0.7085       | 0.6776 | 0.5118 |
> | 10000        | 0.5285   | 0.2799     | 0.0989 | 0.7403       | 0.7006 | 0.5207 |
> | 15000        | 0.5608   | 0.2983     | 0.0847 | 0.7517       | 0.7837 | 0.5535 |
> | 20000        | 0.5495   | 0.3546     | 0.1105 | 0.7469       | 0.8109 | 0.5802 |
> | 25000        | 0.5697   | 0.3828     | 0.1081 | 0.7482       | 0.8070 | 0.5875 |
> | 30000        | 0.5643   | 0.3801     | 0.1387 | 0.7537       | 0.8267 | 0.5937 |
> | 35000        | 0.5592   | 0.3987     | 0.1268 | 0.7409       | 0.8157 | 0.5826 |
> | 40000        | 0.5501   | 0.3805     | 0.1194 | 0.7299       | 0.7996 | 0.5703 |
>
>
> | Number          | MMLU pro | Super GPQA |   HLE  | Health Bench | PubMed | MedQA  |
> |-----------------|----------|------------|--------|--------------|--------|--------|
> | Qwen3-14B-Base  | 0.5844   | 0.2981     | 0.0766 | 0.7535       | 0.8080 | 0.6009 |
> | 5000            | 0.5785   | 0.3047     | 0.0831 | 0.7610       | 0.8613 | 0.6341 |
> | 10000           | 0.5488   | 0.3421     | 0.0896 | 0.7316       | 0.8429 | 0.6536 |
> | 15000           | 0.6032   | 0.3702     | 0.0932 | 0.7524       | 0.8593 | 0.6676 |
> | 20000           | 0.6217   | 0.3548     | 0.1032 | 0.6598       | 0.8938 | 0.6835 |
> | 25000           | 0.6504   | 0.3674     | 0.0966 | 0.6358       | 0.9170 | 0.6898 |
> | 30000           | 0.6686   | 0.3869     | 0.0989 | 0.7489       | 0.9073 | 0.6980 |
> | 35000           | 0.6765   | 0.4096     | 0.1006 | 0.6913       | 0.9183 | 0.7129 |
> | 40000           | 0.6894   | 0.4376     | 0.1096 | 0.6848       | 0.9259 | 0.7354 |
>
>   - For **Qwen3-14B**, we vary the amount of MH synthetic data from 5K to 40K examples, keeping all other settings fixed. We observe a clear upward trend as data volume increases. While individual metrics may fluctuate slightly at intermediate points, the overall pattern is monotonic or near-monotonic improvement with more MH-specific synthetic data. This strongly suggests that the 14B model is indeed in a data-hungry regime in the MH domain.
>
>   - For **Qwen3-8B**, we run the same MH data sweep (5K–40K). We also observe a consistent improvement as data grows, but the gains begin to saturate around 30K examples: beyond ~30K, many MH benchmarks show only marginal or even slightly regressing changes. This contrast indicates that larger models can effectively utilize more MH synthetic data and require a higher data budget before saturation.
>
> **2. Why are MH gains visually less pronounced than HSS for 14B?**
>
> In Fig. 4, all curves are obtained under a **shared training configuration**: for each model size, we use the same RL-optimized prompter (rubric generator) and the same synthetic data generation pipeline, without tuning per-domain or per-scale hyperparameters. Under this unified setup:
>
>   - The MH (Medical & Health) domain starts from a **stronger base model**, as large models typically benefit from richer medical content in pre-training and there exist more MH-related resources overall. As a result, the 14B base model in MH is already relatively close to saturation.
>   - In contrast, the HSS domain exhibits **weaker base performance**, leaving more “headroom” for domain-specialized synthetic data to improve the model.
>
> Consequently, under a fixed fine-tuning budget, the relative improvement appears visually larger in HSS and **more moderate in MH**, even though MH still benefits in a consistent and stable manner across benchmarks.

---

### Author Response · Authors · 2025-12-02
**Global Response**

Dear AC,

Thank you for coordinating the review process. From the authors’ perspective, we would like to briefly report how, in the author response phase, we systematically addressed the core concerns raised by the reviewers and how these changes strengthen the contribution and reliability of the paper.

On the positive side, the reviewers mainly emphasized the following points:

1. **Core insight and framework design `tUAj`, `WCjb`, `a9XS`, `MqSM`.**
   Instead of using heuristic scores, we use *model influence* to guide the selection of rubrics and synthetic data, upgrading traditional static filtering into a closed-loop data generation process driven by optimization feedback. This shifts the focus from “selecting data” to “optimizing the rubric strategy for generating data,” which the reviewers found both novel and well aligned with practical needs. They also appreciated our systematic comparison between the embedding space and the gradient space, noting that many synthetic samples that appear semantically reasonable are not actually beneficial for training, whereas data whose gradients are more aligned with the validation gradient directions are truly useful. This analysis was considered helpful for understanding why existing synthetic data pipelines often fail.

2. **Extensive experiments and cross-domain validation `WCjb`, `MqSM`.**
   Across the HSS and medical domains and under multiple teacher/target model configurations, our rubric-optimization pipeline consistently delivers stable performance gains under different data scales and model capacities.

Regarding limitations and open issues, the comments from the four reviewers can be grouped into the following themes. We conducted targeted analyses and additional experiments for each during the author response phase:

1. **Insufficiently concrete comparison with other SFT/synthetic datasets in terms of statistics `tUAj`, `a9XS`.**
   We systematically compared OptimSyn with several mainstream SFT datasets in terms of number of samples, average length, standard deviation, and vocabulary diversity. The results show that OptimSyn’s statistics lie within the normal range of existing datasets and are not extremely “large/long/rich.” Through correlation analysis, we further demonstrate that neither data scale nor length alone can explain the performance improvements; a more plausible explanation is the improvement of data structure and quality under rubric guidance.

2. **Necessity of using the “influence signal” as feedback `tUAj`, `WCjb`.**
   We systematically compared the influence signal with three representative proxy metrics (QuRating quality score, FineWeb-Edu similarity, and PRRC multi-dimensional quality scores). With the teacher model and training setup fixed and only the feedback signal varied, the influence signal achieves higher average performance than all proxy metrics on six medical benchmarks, and the gains are relatively consistent across benchmarks. This indicates that the influence signal indeed provides additional value and is not simply interchangeable with basic proxies.
   At a higher level, we further compared content-based heuristics, uncertainty-based feedback, and gradient-alignment-based influence signals, emphasizing that the latter directly characterizes “to what extent training on this sample will update parameters in a direction beneficial for the validation set gradient,” thereby aligning more closely with the goal of “selecting the most pedagogically valuable data for a specific downstream task.”

3. **Scalability and computational efficiency `WCjb`, `MqSM`.**
   We explicitly stated the complexity formula of the influence estimation method used in the paper. In practice, we report the actual GPU time under typical settings: computing the influence score for a single sample takes about 0.14 seconds, and for a candidate pool of size 10^5–10^6, the total preprocessing GPU hours are secondary compared with the cost of full SFT training. Moreover, these influence features can be reused across multiple rounds of rubric optimization.

---

> ### Author Response · Authors · 2025-12-02
> **Global Response (Part 2)**
>
> 4. **Dependence on the validation set `MqSM`, `a9XS`.**
>    In the appendix, we describe in detail how the validation set is constructed: we sample from test or development sets of public benchmarks (such as MedQA) that share the same distribution as the target evaluation tasks, and apply a strict deduplication/decontamination procedure to ensure that these samples are used only for gradient and influence estimation and never for SFT training, thereby avoiding leakage.
>    We then systematically studied the effect of validation set size and sampling strategy. With synthetic data and training setup fixed, we gradually increased the validation set size (100–1273 samples) and observed that performance improves steadily with validation size, with clearly diminishing returns after about 800 samples. Even with only 100 or 300 samples, the influence-guided model already significantly outperforms the baseline, although variance is larger under random sampling.
>    To address this, we further compared random sampling, embedding-based diversity sampling, and EvalTree capability-tree-based diversity sampling. We found that the latter two can already approach the performance of using the full validation set with only 100 validation samples, suggesting that the method is not overly sensitive to validation set size as long as semantic and capability coverage are ensured.
>
> We have incorporated all additional experiments and analyses into the revised manuscript. Notably, on November 24, Reviewer `a9XS` expressed agreement with our response and increased their score to 6. Reviewer `WCjb` has also tentatively endorsed our experimental analysis and followed up with further questions regarding the choice of influence function and LoRA space, for which we have provided detailed responses to address all remaining concerns.

---

### Meta-Review · Area_Chair_YZpf · 2026-01-07

**Summary:**

This manuscript proposes a new framework that ueses gradient information to quantify the contribution of synthetic samples and to guide synthetic data generation accordingly. All reviewers appreciate the motivation and empirical performance of the proposed approach. Initially, several concerns were raised, primarily regarding experimental details and scalability, as well as some questions about conceptual depth and potential overlap with existing work.

Following additional experiments and multiple rounds of discussions, one initially negative reviewer shifted to supporting acceptance, while another reviewer expressed satisfaction with both the idea and the experimental results. The remaining concern is that the algorithm design is largely heuristic. The authors provided further discussion to justify their design choices, although this did not receive follow-up feedback from the reviewer.

After carefully reading the paper and the discussion, the AC agrees that the specific choice is indeed heuristic. While such a design is not ideal, it does not constitute a critical flaw when it is strongly supported by empirical evidence.

Overall, the AC recommends ACCEPT.

**Reviewer Concerns:**

This paper received four reviews. Initially, two reviewers held positive opinions, while the other two were negative.

All four reviewers found the core idea appealing, describing it as “interesting” “clear motivation.” Reviewer WCjb (initial score: 4) stated, “I like the closed-loop, feedback-based synthetic optimization view,” and Reviewer a9XS (initial score: 4) considered the overall idea to be novel.

At the same time, reviewers raised multiple questions regarding the experimental setup, mainly focusing on implementation details (e.g., random seeds, test datasets, baseline implementations) and scalability. After the authors provided additional experiments, both initially negative reviewers indicated that they were satisfied with the experimental evaluation.

The remaining discussion with Reviewer WCjb centers on the choice of the space and signal. The AC reviewed both the paper and the discussion and agrees that these choices are primarily motivated by empirical observations rather than theoretical justification. While this is not ideal, it is acceptable for a novel framework, and more principled, theory-backed designs can reasonably be expected in future work.

**Reviewer Scores:**

The initial scores were 4 / 4 / 6 / 6. After the rebuttal, the scores became 4 / 6 / 6 / 6, while the discussion with the remaining critical reviewer was still ongoing.

---

### Decision · Program_Chairs · 2026-01-26

Accept (Poster)